# Earlier ice loss accelerates lake warming in the Northern Hemisphere

Xinyu Li[1], Shushi Peng ®[1] ✉, Yi Xi ®[1], R. Iestyn Woolway ®[2] & Gang Liu[1]

How lake temperatures across large geographic regions are responding to widespread alterations in ice phenology (i.e., the timing of seasonal ice formation and loss) remains unclear. Here, we analyse satellite data and global-scale simulations to investigate the contribution of long-term variations in the seasonality of lake ice to surface water temperature trends across the Northern Hemisphere. Our analysis suggests a widespread excess lake surface warming during the months of ice-off which is, on average, 1.4 times that calculated during the open-water season. This excess warming is influenced predominantly by an 8-day advancement in the average timing of ice break-up from 1979 to 2020. Until the permanent loss of lake ice in the future, excess lake warming may be further amplified due to projected future alterations in lake ice phenology. Excess lake warming will likely alter within-lake physical and biogeochemical processes with numerous implications for lake ecosystems.

More than 90% of the world's lakes are situated north of 30 °N (ref. [1]), and many of these freeze each winter[2]. When lakes freeze, ice acts as a barrier shielding the water surface from atmospheric forcing, notably the incoming radiation, and also acts to increase the lake's albedo[3]. Ice phenology (the timing of ice formation and loss), in turn, regulates the seasonal surface energy and thermal regimes of many lakes worldwide[4,5]. Yet, lake ice is vulnerable to climate change[6,7]. Within a warming world, many lakes have experienced a substantial change in ice phenology, with a later onset and an earlier break-up of ice cover and, subsequently, a longer duration of the ice-free season[6,8,9]. A change in ice phenology can influence the seasonal and inter-annual variability of lake surface water temperature[10,11], with knock-on effects on other physical as well as biogeochemical processes in lakes[12–14].

Historical trends in ice phenology can influence lake warming rates during the months of ice-off and ice-on (i.e., the first and last months of the open-water season, respectively). During the ice-off month (e.g., spring in many north temperate lakes), an earlier break-up of ice cover results in an increase in the amount of incoming short-wave radiation due to reduced surface albedo and an extended period of open-water. This likewise results in an increase in net surface heating, and thus warmer water temperature, with knock-on impacts on lake ecology. For example, warmer conditions during the ice-off

month can facilitate the growth of phytoplankton earlier in the year and, in turn, bring forward the onset of the spring bloom with implications for water quality[15–17]. Warmer conditions during the ice-off month can also lead to an earlier onset of thermal stratification[18] and to warmer surface water temperature during the open-water season[10,11]. The extra heat absorbed by lakes during the open-water season can also contribute to the delay of ice formation in the following winter, with drastic consequences for socioeconomic and cultural ecosystem services[19,20]. Later ice-on dates can, subsequently, lead to more incoming radiation into lakes by extending the open-water season during the ice-on month, but also lead to greater heat loss due to increased evaporation rates[21,22].

Compared to the warming rates often reported during summer, which are influenced by changes in, among other things, air temperature[23], solar radiation[24] and wind speed[25], the magnitude of change in lake surface temperature during the months of ice-off and ice-on remain uncertain. Some in situ observations and model simulations have suggested that earlier ice-off and/or later ice-on in some lakes can lead to excess lake surface warming at these times of the year[26–28] (Fig. 1a). However, this excess lake warming has not yet been explored across larger geographic regions. In this contribution, we aim to fill this knowledge-gap by investigating excess lake warming during

[1]Sino-French Institute for Earth System Science, College of Urban and Environmental Sciences, Peking University, Beijing, China. [2]School of Ocean Sciences, Bangor University, Menai Bridge, Anglesey, UK. ✉e-mail: speng@pku.edu.cn

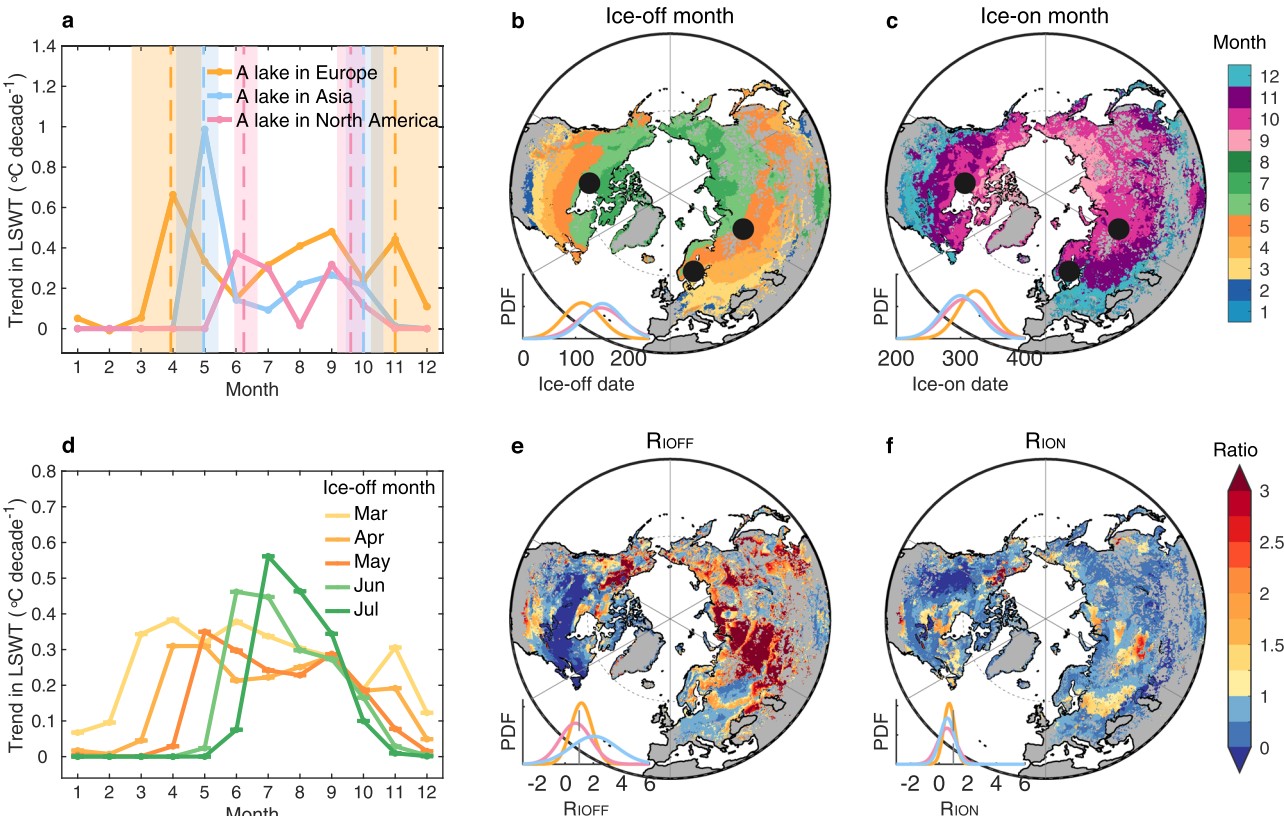

**Fig. 1 | Excess lake warming in the ice-off or ice-on month. a** Trends in monthly lake surface water temperature (LSWT) from 1979 to 2020 across three lakes in Europe, Asia and North America, respectively. The location of the three lakes is shown as black dots in **b**, **c**. The dashed lines and shaded areas represent the mean and range of ice-off date or ice-on date from 1979 to 2020. **b**, **c** Spatial pattern of mean ice-off month (**b**), ice-on month (**c**). **d** Mean trend in monthly LSWT from 1979 to 2020 across lakes with the same ice-off month from March to July. **e**, **f** Spatial pattern of ratios of the trend in LSWT in the ice-off month (LSWT$_{IOFF}$) or LSWT in the ice-on month (LSWT$_{ION}$) to the trend in LSWT during the open-water period ($R_{IOFF}$ (**e**) and $R_{ION}$ (**f**)). The inset in **b**, **c**, **e**, **f** shows the probability density function (PDF) from Europe (orange), North America (pink) and Asia (blue), respectively.

the months of ice-off and ice-on for lakes situated across the Northern Hemisphere (>30 °N). To achieve this aim, we calculated the monthly trends in lake surface water temperature using (i) satellite-observations of 963 globally distributed lakes from 1995 to 2012 (ref. 29); and (ii) modelled lake surface temperatures of 109,405 representative lakes (which represent a 'typical lake' for each 0.25° × 0.25° longitude-latitude grid, referred hereafter simply as lakes; see Methods) from 1979 to 2020, available from the European Centre for Medium Range Weather Forecasts' ERA5 reanalysis product[30]. Note that in our analysis, we independently explored the satellite-derived and modelled data, but only results from the latter are shown below (analyses with the satellite data, which support our findings, are shown in the Supplementary Information).

## Results

### Excess lake warming during the months of ice-off and ice-on

Across the Northern Hemisphere, lake cover duration is longer in high-latitude or high-altitude lakes (Supplementary Fig. 1). Specifically, ice-off dates vary predominantly from March to July (>97% of the lakes analysed) and become progressively later at higher latitudes (Fig. 1b). Ice-on dates mostly occur from September to January (>99%), and are typically later at lower latitudes (Fig. 1c). To illustrate our approach for identifying excess warming across lakes, and to demonstrate the seasonal variation of lake surface temperature trends (1979–2020), we begin our analysis by selecting three ice-covered lakes situated at northern high latitudes (~60 °N) in Europe, Asia and North America and at an elevation of 65 m, 49 m, 164 m, respectively. Our data suggest a peak in the warming trend during the month of ice-off (i.e., when

lakes transition from ice-covered to ice-free) in each of the three lakes (0.4–1.0 °C per decade; Fig. 1a). The calculated trends at this time of year were 1.5–3.1 times those calculated during the open-water season (i.e., averaged over all open-water months). The lake surface temperature trend during the ice-on month was also greater (1.2–1.3 times) than the open-water season in the two lakes located in Europe and North America, but not the one in Asia (Fig. 1a).

When grouping lakes according to the month of ice break-up, our data suggest that the peak warming rate always occurs during the ice-off month, or the subsequent month after ice break-up (Fig. 1d). Moreover, the magnitude of excess warming increases with a later break-up of ice cover (Fig. 1d). To indicate the excess warming during the ice-off/ice-on months, we calculate the ratios of the trend in lake surface temperature during the month of ice-off (LSWT$_{IOFF}$) or ice-on (LSWT$_{ION}$) to the computed trend during the open-water season ($R_{IOFF}$ or $R_{ION}$; see Methods section). With the ice-off month from March to July, the average $R_{IOFF}$ of lakes in the Northern Hemisphere increases from $1.2 \pm 0.9$ to $1.4 \pm 1.3$ ($\pm$ standard deviation across Northern Hemisphere lakes). Our data suggest that 51% of lakes have a $R_{IOFF} > 1$, 34% > 1.5, and 13% > 3 (Fig. 1e). The $R_{IOFF}$ in Asia ($2.0 \pm 1.6$) is much larger than in North America ($0.7 \pm 1.1$) and Europe ($1.2 \pm 0.7$; Fig. 1e). For the ice-on month, the average $R_{ION}$ is $0.6 \pm 0.5$, with the slightly higher $R_{ION}$ in Europe ($0.7 \pm 0.4$) than in Asia ($0.6 \pm 0.5$) and North America ($0.5 \pm 0.6$; Fig. 1f). Only 5% of lakes experience a $R_{ION} > 1.5$, and 1% > 3 (Fig. 1f). Despite covering a shorter time period (1995–2012), satellite observations confirm the excess warming at these times of year, with $R_{IOFF} > 1$ ($R_{ION} > 1$) in 43% (29%) of lakes (Supplementary Fig. 2).

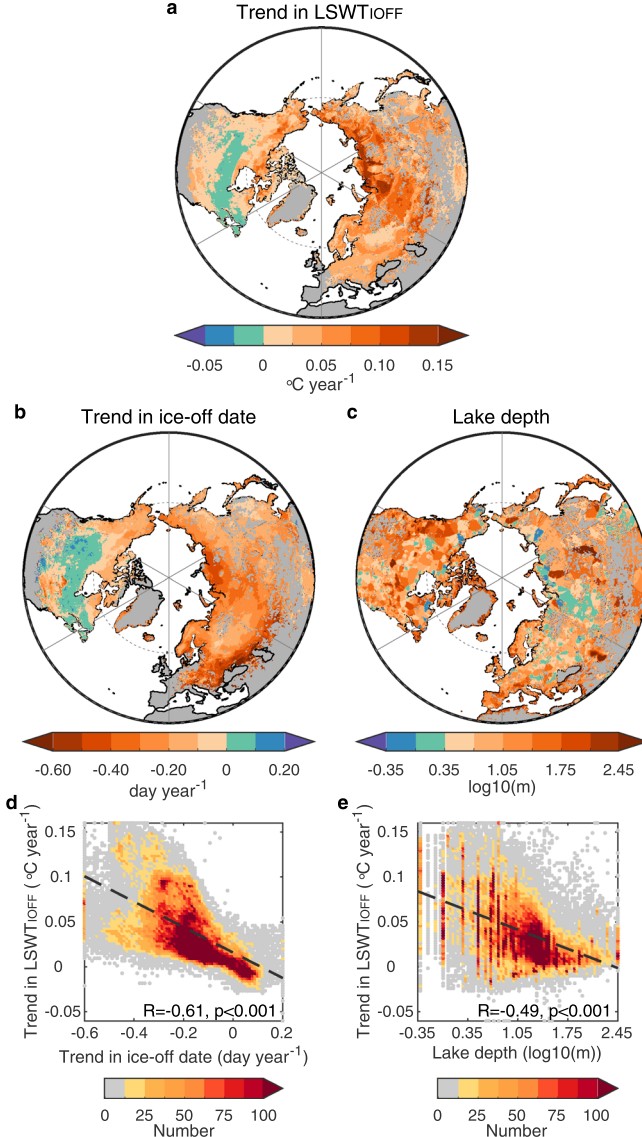

**Fig. 2 | Explanation for excess warming in the ice-off month. a–c** Spatial pattern of the trend in LSWT$_{IOFF}$ (**a**), ice-off date (**b**) and lake depth (**c**). **d, e** Relationship between the trend in LSWT$_{IOFF}$ (*y*-axis) and the trend in ice-off date (**d**) and lake depth (**e**) (*x*-axis). The colour of points in **d**, **e** shows the number of lakes.

## Earlier ice-off and later ice-on result in excess lake warming

As potential drivers of lake surface temperature, climatic variables including surface air temperature[31], downward short-wave radiation[24], and downward long-wave radiation[32], and lake geographic variables including latitude[33] and elevation[34], as well as lake ice phenology[26] may explain the across-lake variations in the magnitude of excess lake warming. However, our analysis suggests that surface air temperature, downward short- and long-wave radiation during the ice-off month only explain 4%, 0% and 5%, respectively, of the spatial variation in $R_{IOFF}$ (Supplementary Fig. 3; Note that the percentages quoted describe the coefficient of determination, which estimates the percentage of variability that can be explained by a regression model and does not imply causality). Moreover, these three variables during the ice-on month only explain 6%, 0% and 5%, respectively, of the spatial variation in $R_{ION}$ (Supplementary Fig. 3). In addition, the across-lake variation in latitude and elevation explains <4% of the spatial variation in $R_{IOFF}$ and $R_{ION}$ (Supplementary Table 1). By contrast, 79% of lakes with maximum monthly warming rates during the month of ice-off experience a significant advancement in the timing of ice break-up ($p < 0.05$), implying

the important role of lake ice phenology on excess lake warming at this time of year.

Our modelled data reports similar spatial patterns between the trends in the date of ice break-up and LSWT$_{IOFF}$ (Fig. 2a, b). For example, in Siberia, many lakes experienced a clear advancement of ice-off date by >0.4 days per year whereas the LSWT$_{IOFF}$ warms at a rate of >0.1 °C year$^{-1}$ (Fig. 2a, b). In southern Canada, the LSWT$_{IOFF}$ data suggests a cooling rate (<−0.01 °C per year) with a delay in ice-off over the study period (<0.05 days per year; Fig. 2a, b). This could be related to a regional cooling in air temperature as well as a diming of short-wave radiation before ice break-up (Supplementary Fig. 4). Across the studied lakes, the trends in ice break-up date correlate positively with the trends in LSWT$_{IOFF}$ ($R = 0.61$, $p < 0.001$; Fig. 2d). In addition to the advancement of ice-off (or earlier ice melting), the seasonality of downward short- and long-wave radiations also contributes to the increase in incoming radiation into lakes. In order to evaluate the net increase in incoming radiation due to these factors, we used the product of the trend in ice-off date by mean downward short- and long-wave radiation during the ice-off month ($\triangle E_{IOFF}^{SW}$ and $\triangle E_{IOFF}^{LW}$, respectively) to indicate extra incoming radiation (Supplementary Fig. 5). We found that the additional influence of absolute radiation in the ice-off month only provides minimal (+4%, +6% for downward short- and long-wave radiation) explanatory power to the across-lake variation in the trend of LSWT$_{IOFF}$ (Supplementary Fig. 5). This is also confirmed by the satellite-derived lake temperatures (Supplementary Fig. 6).

The excess warming observed in the studied lakes is driven primarily by additional radiation input due to the advancement of ice break-up under climate change[10]. However, our analysis also suggests that the magnitude of lake warming could be mediated by lake depth (i.e., the volume of water to be heated prior to the onset of thermal stratification). We hypothesize that lake depth could explain the large variation in the trend of LSWT$_{IOFF}$ across lakes that experience similar changes to the timing of ice-off shown in Fig. 2d. After accounting for the trend in ice break-up, our data suggest that the partial correlation between the trend in LSWT$_{IOFF}$ and lake depth (log-scale) is 0.49 ($p < 0.001$), suggesting an important role of lake depth in mediating excess lake warming. Given that lake stratification will change the volume of water exposed to direct surface heating[35], this hypothesis could be invalid if stratification occurs during the ice-off month. Using the simulated temperature difference between the epilimnion and hypolimnion of lakes (i.e., the surface and bottom layers, respectively), we calculated lake stratification dates across the Northern Hemisphere for 1979–2020 (see Methods section). We found that only 16% of lakes experienced stratification during the ice-off month (Supplementary Fig. 7). Lake depth will also influence the time taken for lakes to stratify following ice break-up. For example, deeper lakes will experience a longer period of deep convective mixing prior to surface waters reaching 4 °C in spring, and in very large and deep lakes, the spring overturn can continue for weeks to months after ice-off[36–38]. Our data suggest an earlier onset date of stratification in 80% of the studied lakes from 1979 to 2020 (Supplementary Fig. 7). This widespread earlier stratification after ice-off month could increase the warming rate of lake surface temperature in the subsequent months[10]. To summarise, a higher proportion of the across-lake variation in LSWT$_{IOFF}$ is explained by the trend in ice-off date (37%) and lake depth (24%) compared to the three climate variables (<11%), thus suggesting a dominant contribution of lake ice phenology and lake properties on excess lake warming during the ice-off month (Supplementary Table 2). Similarly, the ice-on date shows a dominant role in lake surface warming during the ice-on month ($R = 0.63$, $p < 0.001$), but the downward long-wave radiation ($R = 0.42$, $p < 0.001$) and surface air temperature ($R = 0.45$, $p < 0.001$) at this time show a higher correlation with the trend in LSWT$_{ION}$ than lake depth ($R = −0.14$, $p < 0.001$; Fig. 3 and Supplementary Table 2).

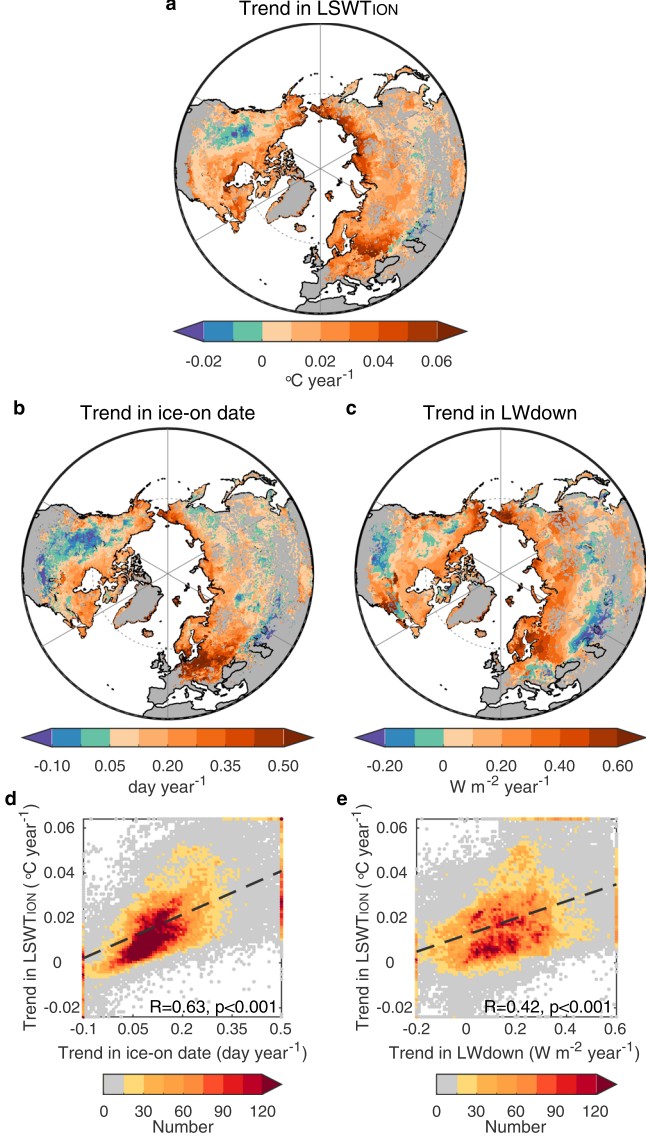

**Fig. 3 | Explanation for excess warming in the ice-on month. a–c** Spatial pattern of the trend in $LSWT_{ION}$ (**a**), ice-on date (**b**) and downward longwave radiation (LWdown) in the ice-on month (**c**). **d, e** Relationship between the trend in $LSWT_{ION}$ (*y*-axis) and the trend in ice-on date (**d**) and downward LWdown (**e**) in the ice-on month (*x*-axis). The colour of points in **d, e** shows the number of lakes.

### Sensitivity of excess lake warming to changes in ice phenology

We calculated the sensitivity of lake warming during the ice-off and ice-on months to changes in ice phenology i.e., what is the magnitude of change in lake surface water temperature with a 1-day advancement of ice-off ($S_{IOFF}$) or a one-day delay of ice-on ($S_{ION}$; see Methods section). The average $S_{IOFF}$ across the North Hemisphere is estimated at 0.14 °C day⁻¹ (Fig. 4a). From 1979 to 2020, the modelled data suggests that the ice-off date has advanced by 8.1 ± 6.4 days, and a corresponding excess warming of lake surface temperature by 1.1 ± 0.9 °C, contributing 81 ± 76% of $LSWT_{IOFF}$ warming (1.8 ± 1.4 °C). Our data suggest that without an advancement to the timing of ice break-up, the $LSWT_{IOFF}$ warming rate would be similar to that calculated during the open water season (1.4 ± 0.7 °C). The average $S_{ION}$ in the North Hemisphere (0.06 °C day⁻¹) is much smaller than the $S_{IOFF}$ (Fig. 4b). The difference between $S_{IOFF}$ and $S_{ION}$ across the studied lakes could be attributed to the much smaller incoming radiation during the ice-on month (256–366 W m⁻²) than during the ice-off month (408–564 W m⁻²; Supplementary Fig. 8). The ice-on date had been delayed by 6.7 ± 6.8 days

from 1979 to 2020, contributing to excess $LSWT_{ION}$ warming by 0.4 ± 0.4 °C, ~64% of the $LSWT_{ION}$ warming trend (0.8 ± 0.7 °C). This excess warming of $LSWT_{ION}$ would be offset by the "dimming" of downward short-wave radiation during the ice-on month, compared to "brightening" during open-water season (Supplementary Table 3 and Fig. 3), resulting in $R_{ION}$ <1 in most grid cells (Fig. 1f).

Given the substantial spatial variation in the $S_{IOFF}$ and $S_{ION}$, we hypothesize that the incoming radiation and the lake depth could regulate the $S_{IOFF}$ and $S_{ION}$ by altering the extra absorbed radiation at the lake surface, as well as the depth of water being heated, with changes in ice phenology. For the $S_{IOFF}$, we found that shallower lakes show a higher sensitivity of $LSWT_{IOFF}$ to ice-off date (Fig. 4c). An increase in lake depth by, for example, an order of magnitude would result in a 60% decrease in $S_{IOFF}$ (~0.12 °C day⁻¹; Fig. 4c). More incoming radiation leads to a higher $S_{IOFF}$, with an increase of ~0.12 °C day⁻¹ in $S_{IOFF}$ in response to an ~100 W m⁻² increase of incoming radiation (Fig. 4e). Although for the $S_{ION}$, the lake depth shows a similar but smaller regulation to the $S_{ION}$, and $S_{ION}$ decreases by 33% (0.03 °C day⁻¹) with increasing lake depth (Fig. 4d). The incoming radiation also positively regulates the $S_{ION}$, but with a smaller magnitude (increase by 0.07 °C day⁻¹ per 100 W m⁻² increase of incoming radiation) compared to the $S_{IOFF}$ (Fig. 4f). In addition, the across-lake variation in elevation explains less than 1% of the spatial variation in $S_{IOFF}$ and $S_{ION}$ (Supplementary Table 1).

## Discussion

Lake ice currently exists in more than half of the 117 million lakes worldwide[9]. As the climate warms, the seasonal timing of ice-off has been well-documented to occur earlier[6,8], with air temperature variations being suggested as a dominant driver of the change in ice phenology[39,40]. In response to a 1.9 °C increase in surface air temperature across the Northern Hemisphere lakes, our modelled data suggests that the timing of ice-off has changed by −8.1 ± 6.4 days from 1979 to 2020. This earlier ice-off date can be explained partly by the seasonality of lake surface warming trends (Supplementary Fig. 9). Between the period 1971–2000 and 2070–2099, lake ice-off dates are projected to occur 15–45 days earlier under Representative Concentration Pathway 8.5 (RCP 8.5) in the Northern Hemisphere[41]. This may lead to a 2.0–6.1 °C extra increase in lake surface temperature during the ice-off month this century according to our findings. Given that more extreme lake ice events, including the occurrence of ice-free winters, are expected with ongoing climate change[42], the amplified increase in lake surface temperature during the ice-off month could be even greater in lakes vulnerable to warmer winters.

The enhancement of lake warming due to earlier ice break-up can last about three months (Supplementary Fig. 10), but the explanation of spatial variation in the lake surface temperature trend by earlier ice-off dates decreases from 37% in the ice-off month to 3% in the 3rd month after ice-off (Supplementary Table 4). The legacy warming effect of earlier ice-off dates on lake water could therefore favour earlier lake stratification[18] (Supplementary Fig. 7). A decoupling between surface and bottom lake water during stratification can reduce the volume of water to be heated and then contribute to a warming of the lake surface[18]. Our modelled data suggests that the onset of stratification has advanced at a rate of 0.2 days per year from 1979 to 2020 across the studied lakes (Supplementary Fig. 7). In all, 16% of lakes experience stratification in the ice-off month, and 82% experience stratification onset in the subsequent 1–3 months. In addition, 14% of lakes experience incomplete overturn (when the number of days between the ice-off date and onset of stratification is <3 days[43]) at least once during the study period (see Methods section). The extended stratification period thereby could contribute to excess lake warming in spring and summer.

Excess lake warming due to earlier ice-off dates could have numerous negative consequences for lake ecosystems. For example,

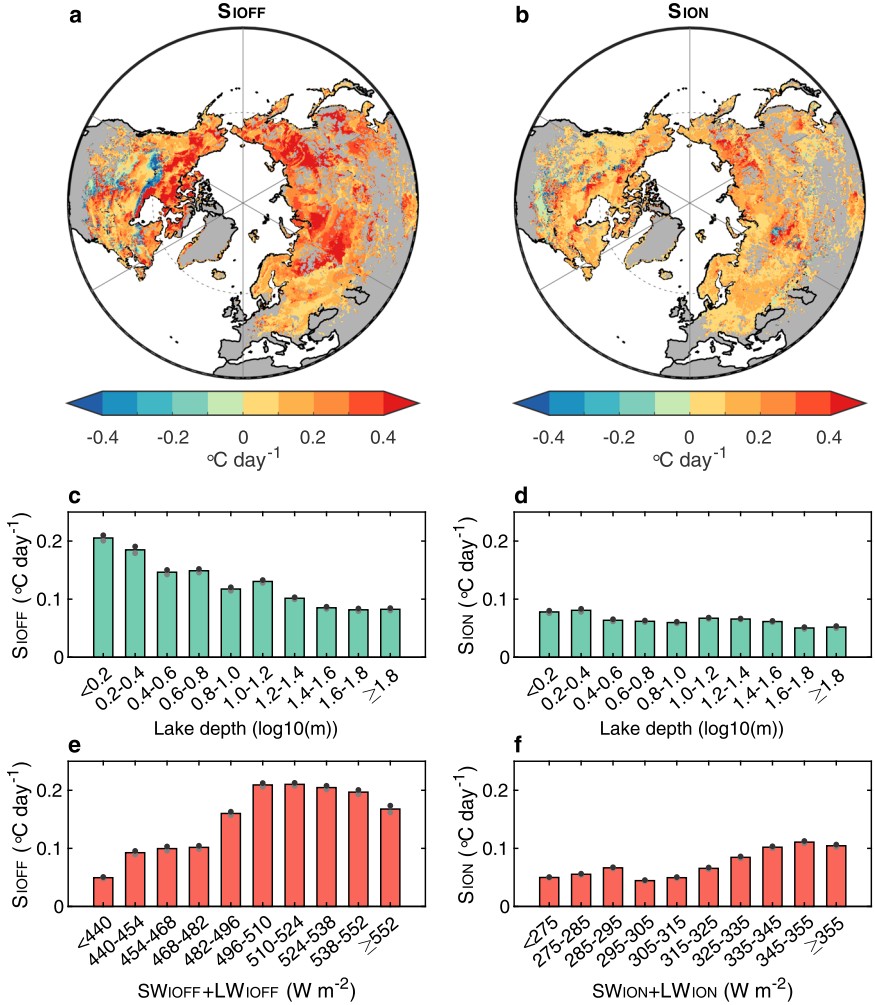

**Fig. 4 | Sensitivity of lake surface temperature to lake ice phenology. a, b** Spatial pattern of the sensitivity of lake surface temperature to changes in ice-off dates ($S_{IOFF}$; **a**) and ice-on dates ($S_{ION}$; **b**). **c, d** Distribution of $S_{IOFF}$ (**c**) and $S_{ION}$ (**d**) as a function of lake depth binned into $0.2 \times \log10$ m intervals. **e, f** Distribution of $S_{IOF}$ and $S_{ION}$ as a function of incoming radiation binned into 14 and 10 W m$^{-2}$ intervals, respectively. The positive values of $S_{IOFF}$ indicate the change in LSWT$_{IOFF}$ in response to the earlier ice-off date while the positive values of $S_{ION}$ indicate the change in LSWT$_{ION}$ due to the later ice-on date. The black and grey dots in **c–f** indicate 95% confidence intervals for the regression fit.

excess lake warming during the month of ice-off could result in the growth of phytoplankton earlier in the year and, in turn, bring forward the onset of the spring bloom[15–17]. The earlier growth of phytoplankton communities has the potential to alter species composition and succession, where increased abundances of early-season taxa or cold-adapted species are favoured[44,45]. Earlier growth may also open-up ecological niches later in the year that enables the growth of potentially harmful filamentous species[44] that may represent a poorer food resource for grazers. Excess lake warming could also deteriorate water clarity by shifting the dominant communities from phytoplankton to cyanobacteria[46–48], since the warmer water temperature could favour growth of the latter[49,50]. On the other hand, excess lake warming due to earlier ice break-up could lead to stronger thermal stratification, resulting in a reduced supply of nutrient rich bottom waters to the near-surface layer, as previously reported in both marine[51–54] and lacustrine systems[55–58]. Moreover, stronger stratification could result in lake deoxygenation at depth[59]. The oxygen-sensitive fishes in deep and cold water could thereby face a risk of habitat loss and die-offs[60–62].

Although our results are robust and could bridge a critical knowledge-gap, some limitations should be considered when interpreting our key findings. Notably, our analysis is based on lake surface water temperature derived from modelled and satellite data, each of which provides an approximation of lake thermal responses to climate change. For example, satellite data represent the lake temperature of skin-surface, a layer with a thickness <0.1 mm, from which thermal radiation is emitted by the lake. Thus, the satellite data may differ from bulk temperature measured at the depth of a few centimeters by a thermometer[63]. Typically, the difference in lake water temperature between skin-surface and bulk layers is a few tenths of a degree, but depends on the lake conditions. Although the skin effect is variable, the satellite lake surface temperature is nonetheless tightly coupled to the lake surface temperature as measured conventionally, particularly over long timescales, and has been used extensively in previous studies quantifying worldwide aspects of lake thermal dynamics[64–67]. To maximise the number of observations available for each of the studied lakes, we also investigated the lake-mean surface temperature as opposed to, for example, retrievals from the lake centre pixel[68]. The data used thus represent the average temperature across the surface area of each lake. In turn, the intra-lake heterogeneity of ice and surface water temperature responses to climate change[65,69] were not considered in this study. This can be particularly important for large lakes where the time taken for deeper central regions to warm during spring and summer, and to cool in autumn and winter, is substantially different from the shallow nearshore regions[70–72].

To correspond with the lake-mean satellite observations described above, we elected to investigate simulated water temperatures

from a one-dimensional (1D) lake model. Compared to the more comprehensive, but computationally expensive, 2D or 3D models, a 1D model assumes a simple bathymetry for the simulated lakes, thus introducing an additional source of uncertainty in our projections[73–75]. Furthermore, our simulations do not consider temporal changes in water clarity (via the light attenuation coefficient, $K_d$). Water clarity can influence both lake surface and bottom water temperature, as darker lakes typically absorb more incoming radiation at the surface, which can lead to (1) an increase in surface water temperature in summer but result in faster cooling in autumn[76,77], and (2) a decrease in summer bottom temperature due to thermal shielding[78,79]. In addition, as there is no water balance equation in FLake, lake depth and surface area are constant in time, which can be an additional source of uncertainty, particularly in some shallow lakes where lake levels change dramatically at seasonal and inter-annual timescales. Despite these limitations, we stress that this model has been used previously to successfully estimate the thermal environment of lakes globally, and our results provide an important step forward in understanding changes in lake thermal conditions within a warming world.

Both ice phenology and lake surface temperature are essential physical lake variables, which are particularly vulnerable to climate change[22]. Changes in lake ice phenology, notably a later onset and earlier break-up of ice cover, have been suggested to occur in lakes worldwide and are expected to continue with future warming[8,9,80,81]. In this study, we calculate excess lake warming during the months of ice-off and ice-on due to an earlier loss of ice cover and later ice formation throughout the Northern Hemisphere. Meanwhile, as a result of higher incoming solar radiation during the ice-off months, the sensitivity of excess warming to the timing of ice loss is quantitatively higher than during the month of ice-on. Our results highlight the excess warming during these transitional months between the ice and open-water seasons using historical datasets. Projecting future lake warming due to changes in ice phenology is important for understanding lake thermal regimes, lake ecological processes and the provision of lake ecosystem services within a warming world.

## Methods
### ERA5 reanalysis data
To calculate monthly lake warming rates across the Northern Hemisphere (1979–2020), we used hourly lake surface water temperature (notably the temperature of the upper mixed layer) simulations from European Centre for Medium Range Weather Forecasts (ECMWF) ERA5, which has a spatial resolution of 0.25° × 0.25° (refs. 30,82). Lake surface water temperature in ERA5 is simulated with the FLake model[83–85], which is coupled into the Hydrology Tiled ECMWF Scheme for Surface Exchanges over Land (HTESSL) of integrated forecast system (IFS). ERA5 lake surface temperatures have been validated against observations in previous studies[86], and have been used to simulate lake thermal responses to climate change[18,86]. The lake grid cells in ERA5 are identified with a >0% lake cover fraction. For lake warming attribution, hourly surface solar radiation downwards, surface thermal radiation downward, surface 2-metre air temperature, lake bottom temperature, and static lake depth with a spatial resolution of 0.25° × 0.25° were also downloaded from the ERA5 reanalysis datasets for 1979–2020. All hourly climate and lake temperature data were aggregated into daily or monthly averages for further analyses.

### Satellite observations
We investigated satellite-derived lake surface water temperature observations from the Along Track Scanning Radiometer (ATSR) Reprocessing for Climate: Lake Surface Water Temperature and Ice Cover (ARC-Lake) dataset (www.laketemp.net)[29,87] to calculate warming rates across the Northern Hemisphere lakes. ARC-Lake provides daily lake-wide mean surface temperature in 1628 lakes from June 1995 to April 2012, produced by ATSR-2 and the Advanced ATSR. Lake-mean

satellite observations for 963 lakes north of 30 °N are used in this study to average across the intra-lake heterogeneity of lake thermal responses to climate change[65], and to be comparable to the lake mean model used. Lake warming rates show good agreement between ERA5 and ARC-Lake (Supplementary Fig. 11).

A separate lake surface temperature product, version 1.0 of European Space Agency's Climate Change Initiative (CCI) Lakes project (CCI Lakes; http://cci.esa.int/lakes)[88] was also used to validate the results from ERA5 (Supplementary Fig. 12). This product provides daily lake surface water temperature for 250 lakes worldwide with a spatial resolution of 1 km × 1 km from July 1996 to December 2019. We only used lake temperature data from CCI Lakes from 2007 to 2019 in this study given the temporal coverage is <20% before 2007 across lake grid cells in the Northern Hemisphere, and only lake grid cells with available data for >2 days in a month (available month) and >9 available months in a year were used to calculate the monthly trends in lake surface water temperature. For comparison with ERA5, lake surface temperatures from CCI Lakes were aggregated to a 0.25° longitude-latitude resolution. The lake warming rates calculated from ERA5 and CCI Lakes also show a good agreement (Supplementary Fig. 12), but further analysis can't be conducted due to the limited spatial coverage of CCI Lakes. Owing to the relatively shorter temporal coverage for ARC-Lake (1995–2012) compared to ERA5 (1979–2020), we only showed the results from ERA5 in the main text while the results from ARC-Lake and CCI Lakes in the Supplementary.

### Ice phenology data
Lake ice-on date is defined as the first date when the lake is totally ice-covered, while ice-off date is defined as the first date when the lake is totally ice-free[6,89]. To calculate annual ice phenology of lakes across the Northern Hemisphere, we used daily lake surface water temperature from ERA5 during the period 1979–2020 and from ARC-Lake during the period 1995–2012. For each lake grid cell (ERA5) or individual lake (ARC-Lake), we followed Layden et al.[87] and identified ice-cover periods when lake surface temperature is <1 °C. This method has been used in previous studies[66]. The ice-on date was defined as the first day when ice-cover periods last for 10 consecutive days after 1 July while the ice-off date was the first day when ice-free periods last for 10 consecutive days. This 10-day threshold was used to account for periods of intermittent ice cover during the transitional periods of ice cover and open-water. We only selected lake grid cells (ERA5) or individual lakes (ARC-Lake) with at least 1-year ice phenology data for further analysis. Meanwhile, we excluded lakes with an ice-cover duration <30 days to prevent lakes from experiencing break-up and freeze-up in the same month (this included <0.2% of lakes in ARC-Lake and <0.3% of lake grid cells in ERA5). To validate the annual ice phenology data derived from daily LSWT from ERA5 (1979–2020) and ARC-Lake (1995–2012), we also used a dataset of lake ice phenology from Advanced Microwave Scanning Radiometer for EOS and Advanced Microwave Scanning Radiometer 2 (AMSR-E/2) sensors[90,91]. This dataset contains daily ice phenology time series for 76,671 lake pixels with a spatial resolution of 5 km × 5 km across the Northern Hemisphere from 2002 to 2015, derived from brightness temperature data of AMSR-E/2 sensors. We selected lake pixels with temporal coverage >85% of the period 2002–2015. Annual ice-off and ice-on dates derived from ERA5, ARC-Lake and AMSR-E/2 show a good agreement ($R > 0.73$) in lake grid cells across the Northern Hemisphere (Supplementary Figs. 13–15). To test the robustness of lake ice phenology calculated using lake surface temperature from ERA5 in the main text, we also used lake ice thickness from ERA5 to calculate annual ice phenology. The ice-cover periods of each lake grid cell were identified when daily ice thickness is >0.001 m. A good agreement is found in annual ice-off and ice-on dates between the algorithm using lake ice thickness and that using lake surface temperature ($R > 0.95$; Supplementary Fig. 16). However, we also found earlier annual ice-on dates and later annual ice-off dates

derived from ERA5 lake ice thickness than that from AMSR-E/2 and that from ERA5 lake surface temperature (Supplementary Figs. 16 and 17). This may be related to missing or under-represented ice-related processes in FLake, e.g., the absence of the snow module in FLake, while snow cover could have higher albedo and greater insulation effects on ice formation or breakup[39]. The later annual ice-off dates may also be related to the absence of heating of water by solar radiation penetrating down the ice cover[26]. Similar spatial patterns of trends in ice-on and ice-off dates using the two algorithms are also found in the Northern Hemisphere (Supplementary Fig. 18), illustrating the robustness of our results.

### Impact of ice phenological trends on lake warming trends

We used the ratio of the trend in $LSWT_{IOFF}$ and $LSWT_{ION}$ to the lake warming trend during the open-water season ($R_{IOFF}$ and $R_{ION}$) to indicate excess warming in the ice-off and ice-on month across the Northern Hemisphere, respectively. To indicate the sensitivity of lake warming during the ice-off and ice-on months to changes in ice phenology, we calculated the ratio of the trend in $LSWT_{IOFF}$ to the trend in ice-off dates ($S_{IOFF}$), and the ratio of the trend in $LSWT_{ION}$ to the trend in ice-on dates ($S_{ION}$). The trends in $LSWT_{IOFF}$ and $LSWT_{ION}$ are trends of monthly lake surface water temperature for the months of ice-off and ice-on from 1979 to 2020, respectively. In this way, the open-water season of each lake grid cell was defined as the months after the ice-off month and before the ice-on month. The ice-off (ice-on) month for each lake was defined as the month of mean ice-off (ice-on) date during the period 1979–2020, not varying between years. Given that when ice-off occurs at the end of a month the ice-off has a limited impact on lake surface water temperature in that month but a larger impact during the following month, we also tested another definition for ice-off and ice-on month. When the mean ice-off date is later than the 25th day of the month or the mean ice-on date is earlier than the 5th day of the month, then the following month is defined as the ice-off month and the previous month is the ice-on month. With this alternative definition, we found similar results that lake ice phenology is the dominant factor of excess lake warming in both ice-on and ice-off month (Supplementary Fig. 19).

We used correlation analysis to explain the impacts of climate variables, lake ice phenology and lake depth on the excess lake warming during the ice-on and ice-off month. In addition, we also consider the role of lake stratification after ice-off on lake warming. On the basis of the methods described by Woolway et al.[92], we used the difference between lake surface and bottom temperature from ERA5 to calculate the lake stratification during 1979–2020. The onset of stratification is defined as the first day when the temperature difference between the epilimnion and the hypolimnion is >1 °C and lasts for 3 consecutive days (Supplementary Fig. 20). The occurrence of incomplete overturn of a lake is defined as when the lake's mixing duration (the number of days between the ice-off date and onset of stratification) is <3 days[43].

### Data availability

All observations and modelled data that support the findings of this study are available as follows. ERA5 data used in this study are available from https://cds.climate.copernicus.eu/cdsapp#!/dataset/reanalysis-era5-single-levels?tab=overview. The satellite lake temperature products from ARC-Lake and CCI Lakes are obtained from www.laketemp.net and http://cci.esa.int/lakes, respectively. Satellite ice phenology data from AMER-E/2 are available at https://nsidc.org/data/NSIDC-0726/versions/1.

### Code availability

The analyses were performed using MATLAB (R2020a). All computer codes for the analysis of the data are available from the corresponding author on reasonable request.

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

## Acknowledgements

R.I.W. was supported by a UKRI Natural Environment Research Council (NERC) Independent Research Fellowship (grant number NE/T011246/1). We thank the European Space Agency Climate Change Initiative project for providing the satellite data. The study was supported by the National Natural Science Foundation of China (grant numbers 41830643 and 41722101). We thank the Copernicus Climate Change Service for their provision of publicly available ERA5 hourly data. We are also grateful for the computational resources provided by the High-performance Computing Platform of Peking University's supercomputing facility.

## Author contributions

S.P. designed the study. X.L. performed the analysis and created all the figures. S.P., X.L., Y.X., I.R.W. and G.L. drafted, commented and wrote the manuscript.

## Competing interests

The authors declare no competing interests.
