## [Peer Review File · Nature Communications]

Earlier ice loss accelerates lake warming in the Northern HemisphereReviewers' Comments:

Reviewer #1:

Remarks to the Author:

Earlier ice loss accelerates lake warming in the Northern Hemisphere

Comments to the authors:

Global warming driven changes in lakes' thermal regimes, particularly surface water warming, have been recognised, investigated, and broadly reported for Northern inland water bodies.

The problem investigated in this manuscript is significant both scientifically and societal. Previous studies have shown global trends of lakes' surface temperature (LST) and expected ice loss in future scenarios. Thus, in contrast to previous works, the present manuscript dives deeper into examining the processes involved in the temperature rise and loss of the ice-on season in lakes.

In this manuscript, the authors analyse and report how the temperature of a large sample of Northern lakes is being modified by the changes in ice phenology, i.e., the timing of seasonal ice formation and loss. The analysis combines satellite data and simulations—of lakes' temperature evolution—for mid to high latitude Northern lakes to quantify variability and trends on the ice-on period and its relationship with surface water temperature. The above analysis is framed on a four decades time window. The paper proposes that the excess warming found in the studied lakes is due to the shortening of the ice-on period, which leads to an excess of radiative heat input in the water bodies that remains and affects the global evolution of the LST. The latter result is expected since surface waters are exposed to direct radiative heating for longer periods. However, in contrast to previous studies, the authors provide warming rates and variability for LST due to the pervasive ice loss. More significantly, the authors quantify the relative relevance of climate variables, ice loss and mean depth in the warming rate of lakes' surface waters during ice-off periods. A remarkable result is that ice-loss and lake mean depth explain 37% and 24% of the spatial variation in excess lake surface warming, leaving climate variables as secondary drivers for the LST.

My main criticism is that there is no discussion about the methods' limitations—both the satellite data and modelling framework used in this work deserves a paragraph in the discussion section. In the current stage of the manuscript, results rely entirely on a previously used framework, which was validated to derive specific information of the lakes' temperature field evolution. However, since this work attempts to untangle processes of much higher complexity than those previously reported, I think the readership deserves the caveat and scopes of the methods here adopted. Therefore, it is relevant that the warming rates found in this work are more significant than the variability and inherent numerical modelling's misfit resulting from not integrating, for instance, topography effects that lead to deviations between one-dimensional and multi-dimensional models and the sediment heat flux that can affect thermal regimes during the ice-on season. One might think that conclusions would not change if more comprehensive modelling approaches were adopted. Still, it is relevant to stress the physical framework that leads to the reported results. The latter would help to assess the reliability of the results and define future directions we should take to revisit the research question posed in this work and underpin its outcomes.

The following three references may help the authors to discuss how topography, latitude and pre and under-ice thermal regimes can affect the effective warming rate of ice-covered water bodies:

Ulloa H. N, Winters, B. Wüest, A. and Bouffard, D. (2019). Differential Heating Drives Downslope Flows that Accelerate Mixed-Layer Warming in Ice-Covered Waters. *Geophysical Research Letters* <https://doi.org/10.1029/2019GL085258>

Yang et al. (2020). A New Thermal Categorization of Ice-Covered Lakes.

Ramón, C. L., Ulloa, H. N., Doda, T., Winters, K. B., and Bouffard, D.: Bathymetry and latitude modify lake warming under ice, *Hydrol. Earth Syst. Sci.*, 25, 1813–1825, <https://doi.org/10.5194/hess-25-1813-2021>, 2021.

Reviewer #2:

Remarks to the Author:

Summary:

This paper analyzed lake ice cover and lake water temperature data from the ERA-5 reanalysis and ARC-Lake satellite data to evaluate the trends in ice on and off dates in lakes in the northern hemisphere, as well as the trends in lake surface temperature during the ice on & off months. The authors claim that the warming of lake surface temperature was predominant in the ice-off months compared to that in the ice-on months, that earlier ice off resulted in earlier onset of stratification, that the warming was slower in deeper lakes, and that the predominant warming in the ice-off months could be well explained by earlier ice-off rather than incoming radiations or increasing air temperature.

Major comments:

Significance

1. While the topic is important and has significant impacts on subsequent processes such as lake biogeochemistry. Most of the authors' findings are not new. Earlier onset of stratification and resulting excess warming in lakes are well documented in Woolway et al. (202), Austin & Coleman (2008, Lake Superior only), as well as loss of ice cover across the lakes in the northern hemisphere (Sharma et al. 2019), and loss of ice cover in the Great Lakes (Wang et al., 2012; 2018) The slower warming in deeper lakes makes sense and is not a whole new finding.

2. I felt that these findings around the lake surface warming in responding to ice cover loss provide only a limited snapshot of the much bigger complex picture of the lake-ice systems. The ice loss and the excess warming of the lake surface are intermediary processes. As individual processes, they are relatively well understood. The more important yet unknown is how the lake-ice systems respond to the climate forcing (air temperature, radiation, snowfall on the ice) through all these intermediary processes. I think the author will need to re-work their analyses in this context. For example, the authors could possibly discuss the frequency of warm winters when lakes were ice free, occurrence of incomplete overturns in dimictic and monomictic lakes, and how they impacted summer stratification overall. Another possibility is to examine a tipping point (if any), such as lake regime change from dimictic to monomictic (e.g., Woolway et al 2020) and cumulative freezing degree days over each lake at which ice cover presented a flexion point in its decrease. They are just suggestions to consider.

3. I am slightly surprised that the paper does not offer any insight on how ice duration and excess warming are related to the elevations, sizes, and latitudes of the lakes. Aren't they any significant relationships?

4. The slight cooling of the lakes in southern Canada is interesting, but the reason for this is not discussed at all.

Validity

5. The methodology used in the paper is mostly valid, but there are some insufficient descriptions on

their analyses. For example, the authors use 'air temperature' and other surface meteorological variables to examine their relationship with R_ION, R_IOFF etc but there is no description of what period of air temperature etc was used (e.g., annual mean, winter value, or ice-on/off months?). Some of them are pointed out in 'other points' below.

6. I am concerned with the author's statement in the abstract that 'Notably, the calculated trend in the date of ice loss explains 37% of the spatial variation in excess lake warming, which is greater than that explained by air temperature and downward short and long-wave radiation (< 11%)'. Similar expressions were found multiple times throughout the manuscript. Because ice melting and breakups are driven by incoming radiation and warm air above, they are not independent forcings, so the statement could be misleading.

Other points:

L15: "in the seasonality of lake ice to surface temperature". Please specify surface air temperature or surface water/ice temperature.

L24: "Excess lake warming during autumn/winter is also sensitive to the date of ice formation, but with a weaker magnitude (0.06 C day⁻¹) due to less incoming radiation." This sentence seems to imply a cause-and-result relationship but it is unlikely that ice formation timing resulted in warming in preceding autumn/winter.

L26: "Until the permanent loss of lake ice this century, excess lake warming may be further amplified due to projected future alterations in lake ice phenology." Ice loss is an important factor but another critical point is when warming reaches the point where overturns no longer occur. I think it's worth noting this point.

L77: Fig. 1b is referred before Fig. 1a.

L81: Unsure why the three selected lakes are representative.... why not means of more lakes over the region?

L85: "The lake surface temperature trend... (Fig. 1a)". It seems inadequate to draw conclusions for the entire NA, Europe or Asia based on the 3 lakes only.

L94: For R_ION, R_IOFF calculation, how were the open water seasons defined? unique to each lake and year or constant across the lakes and the study period?

L132: "We propose that the excess warming observed in the studied lakes is driven primarily by additional radiation input due to the advancement of ice break-up under climate change." This is a well-known ice-albedo feedforward process and not really a new proposal.

L172: "The difference between SIOFF and SION across the..." Also water temperature has a lower limit at the freezing degree, so below this point, it is insensitive no matter how cold the air above is.

ERA-5 lake data: What's the minimum size of lakes covered in this study? If more than one lake is included in a 0.25degx0.25deg cell, are they considered as one lake consolidated or is there any subgrid scale treatments?

L300: "identified ice-cover periods when lake surface temperature is less than 1 C". Why do you use the lake surface temperature to determine an ice-covered period when you have ice cover data? Why

not use the ice cover data directly?

L306: "Meanwhile, we excluded lakes with an ice-cover duration < 30 days to prevent lakes experiencing break-up and freeze-up in the same month (< 0.2%)." What's the fraction of data removed by this criteria?

Calculation methods of S_ION, S_IOFF, the product of the trend in ice-off date by mean downward short- and long-wave radiation during the ice-off month are not described in Methods. For S_ION & S_IOFF, the main text says "See Methods".

Figure 2: This figure appears very crowded. b.c.g.h are really hard to see. I understand that there is a limitation in the figure numbers. However, I encourage the authors to make an effort to make it more concise.

Response to the reviewers

To Reviewer #1

Reviewer #1 General comments

Global warming driven changes in lakes' thermal regimes, particularly surface water warming, have been recognised, investigated, and broadly reported for Northern inland water bodies.

The problem investigated in this manuscript is significant both scientifically and societal. Previous studies have shown global trends of lakes' surface temperature (LST) and expected ice loss in future scenarios. Thus, in contrast to previous works, the present manuscript dives deeper into examining the processes involved in the temperature rise and loss of the ice-on season in lakes.

In this manuscript, the authors analyse and report how the temperature of a large sample of Northern lakes is being modified by the changes in ice phenology, i.e., the timing of seasonal ice formation and loss. The analysis combines satellite data and simulations—of lakes' temperature evolution—for mid to high latitude Northern lakes to quantify variability and trends on the ice-on period and its relationship with surface water temperature. The above analysis is framed on a four decades time window. The paper proposes that the excess warming found in the studied lakes is due to the shortening of the ice-on period, which leads to an excess of radiative heat input in the water bodies that remains and affects the global evolution of the LST. The latter result is expected since surface waters are exposed to direct radiative heating for longer periods. However, in contrast to previous studies, the authors provide warming rates and variability for LST due to the pervasive ice loss.

More significantly, the authors quantify the relative relevance of climate variables, ice loss and mean depth in the warming rate of lakes' surface waters during ice-off periods. A remarkable result is that ice-loss and lake mean depth explain 37% and 24% of the spatial variation in excess lake surface warming, leaving climate variables as secondary drivers for the LST.

[Response] We thank the reviewer for highlighting the importance of our study and for their positive feedback.

[Reviewer #1 General Comment 1]

My main criticism is that there is no discussion about the methods' limitations—both the satellite data and modelling framework used in this work deserves a paragraph in the discussion section. In the current stage of the manuscript, results rely entirely on a previously used framework, which was validated to derive specific information of the lakes' temperature field evolution. However, since this work attempts to untangle processes of much higher complexity than those previously reported, I think the readership deserves the caveat and scopes of the methods here adopted. Therefore, it is relevant that the warming rates found in this work are more significant than the variability and inherent numerical modelling's misfit resulting from not integrating, for instance, topography effects that lead to deviations between one-dimensional and multi-dimensional models and the sediment heat flux that can affect thermal regimes during the ice-on season. One might think that conclusions would not change if more comprehensive modelling approaches were adopted. Still, it is relevant to stress the physical framework that leads to the reported results. The latter would help to assess the reliability of the results and define future directions we should take to revisit the research question posed in this work and underpin its outcomes.

The following three references may help the authors to discuss how topography, latitude and

pre and under-ice thermal regimes can affect the effective warming rate of ice-covered water bodies:

Ulloa H. N., Winters, B. Wüest, A. and Bouffard, D. (2019). Differential Heating Drives Downslope Flows that Accelerate Mixed-Layer Warming in Ice-Covered Waters. Geophysical Research Letters

<https://doi.org/10.1029/2019GL085258>

Yang et al. (2020). A New Thermal Categorization of Ice-Covered Lakes.

Geophysical Research Letters

<https://doi.org/10.1029/2020GL091374>

Ramón, C. L., Ulloa, H. N., Doda, T., Winters, K. B., and Bouffard, D.: Bathymetry and latitude modify lake warming under ice, Hydrol. Earth Syst. Sci., 25, 1813–1825, <https://doi.org/10.5194/hess-25-1813-2021>, 2021.

[Response] We thank the reviewer for the valuable comments and suggestions. This is indeed a useful comment. We agree with the reviewer that the previously used modelled data from one-dimensional model FLake and satellite data from ARC-Lake might limit our results in this study and that these limitations should be carefully described. Following the reviewer’s suggestions, we added two paragraphs in the revised version to discuss the limitation of using these modelled and satellite data (Lines 240-276, also copied as below). We also now cite the papers mentioned by the reviewer.

“Although we consider our results robust, and believe that they bridge a critical knowledge-gap, there are some limitations to consider when interpreting our key findings. Notably, our study is based on an analysis of modelled and satellite-derived lake surface

water temperature data, each of which provides an approximation of lake thermal responses to climate change. For example, satellite-derived lake surface temperatures represent the skin temperature of water, which is the temperature of a layer < 0.1 mm thick from which thermal radiation is emitted by the lake. Thus, the satellite data may differ from the temperature as measured by a thermometer a few centimeters below the water surface (Wilson et al., 2013). Typically, the temperature difference between skin and sub-skin lake surface temperature is a few tenths of a degree, but depends on the lake conditions. Although the skin effect is variable, the satellite lake surface temperature is nonetheless tightly coupled to the lake surface temperature as measured conventionally, particularly over long timescales, and has been used extensively in previous studies quantifying worldwide aspects of lake thermal dynamics (Maberly et al., 2020; Woolway and Merchant 2018; Woolway and Merchant 2019; Fichot et al., 2019). To maximize the number of observations available for each of the studied lakes, we also investigated the lake-mean surface temperature as opposed to, for example, retrievals from the lake center pixel (Schneider and Hook 2010). The data used thus represent the average temperature across the surface area of each lake. In turn, the intra-lake heterogeneity of ice and surface water temperature responses to climate change (Mason et al., 2016; Woolway and Merchant 2018) were not considered in this study. This can be particularly important for large lakes where the time taken for deeper central regions to warm during spring and summer, and to cool in autumn and winter, is substantially different to the shallow nearshore regions.

To correspond with the lake-mean satellite observations described above, we elected to investigate simulated temperatures from a one-dimensional (1D) lake model, which also largely represents lake-mean conditions. Compared to the more comprehensive, but

computationally expensive, 2D or 3D models, a 1D model assumes a simple bathymetry for the simulated lakes, thus introducing an additional source of uncertainty in our projections (Ramon et al., 2021; Yang et al., 2021; Ulloa et al., 2019). Furthermore, our simulations do not consider temporal changes in water clarity (via the light attenuation coefficient, K_d). Water clarity can influence both lake surface and bottom water temperature, as darker lakes typically absorb more incoming radiation at the surface, which can lead to 1) an increase in surface water temperature in summer but result in faster cooling in autumn (Rinke et al., 2010; Heiskanen et al., 2015), and 2) a decrease in summer bottom temperature due to thermal shielding (Rose et al., 2016; Bartosiewicz et al., 2019). In addition, as there is no water balance equation in FLake, lake depth and surface area are constant in time, which can be an additional source of uncertainty, particularly in some shallow lakes where lake levels change dramatically at seasonal and inter-annual timescales. Despite these limitations, we stress that this model has been used previously to estimate successfully the thermal environment of lakes globally, and our results provide an important step forward in understanding changes in lake thermal conditions within a warming world.”

References

- Bartosiewicz, M., et al. (2019) Hot tops, cold bottoms: Synergistic climate warming and shielding effects increase carbon burial in lakes. *LIMNOLOGY AND OCEANOGRAPHY LETTERS*, 4, 132-144.
- Fichot, C. G., et al. (2019) Assessing change in the overturning behavior of the Laurentian Great Lakes using remotely sensed lake surface water temperatures. *REMOTE SENSING OF ENVIRONMENT*, 235.

- Heiskanen, J. J., et al. (2015) Effects of water clarity on lake stratification and lake-atmosphere heat exchange. *JOURNAL OF GEOPHYSICAL RESEARCH-ATMOSPHERES*, 120, 7412-7428.
- Maberly, S. C., et al. (2020) Global lake thermal regions shift under climate change. *NATURE COMMUNICATIONS*, 11.
- Mason, L. A., et al. (2016) Fine-scale spatial variation in ice cover and surface temperature trends across the surface of the Laurentian Great Lakes. *CLIMATIC CHANGE*, 138, 71-83.
- Ramon, C. L., et al. (2021) Bathymetry and latitude modify lake warming under ice. *HYDROLOGY AND EARTH SYSTEM SCIENCES*, 25, 1813-1825.
- Rinke, K., et al. (2010) A simulation study of the feedback of phytoplankton on thermal structure via light extinction. *FRESHWATER BIOLOGY*, 55, 1674-1693.
- Rose, K. C., et al. (2016) Climate-induced warming of lakes can be either amplified or suppressed by trends in water clarity. *Limnology and Oceanography Letters*, 1, 44-53.
- Schneider, P. & S. J. Hook (2010) Space observations of inland water bodies show rapid surface warming since 1985. *GEOPHYSICAL RESEARCH LETTERS*, 37.
- Ulloa, H. N., et al. (2019) Differential Heating Drives Downslope Flows that Accelerate Mixed-Layer Warming in Ice-Covered Waters. *GEOPHYSICAL RESEARCH LETTERS*, 46, 13872-13882.
- Wilson, R. C., et al. (2013) Skin and bulk temperature difference at Lake Tahoe: A case study on lake skin effect. *JOURNAL OF GEOPHYSICAL RESEARCH-ATMOSPHERES*, 118, 10332-10346.

Woolway, R. I. & C. J. Merchant (2018) Intralake Heterogeneity of Thermal Responses to Climate Change: A Study of Large Northern Hemisphere Lakes. JOURNAL OF GEOPHYSICAL RESEARCH-ATMOSPHERES, 123, 3087-3098.

Woolway, R. I. & C. J. Merchant (2019) Worldwide alteration of lake mixing regimes in response to climate change. NATURE GEOSCIENCE, 12, 271-+.

Yang, B., et al. (2021) A New Thermal Categorization of Ice-Covered Lakes. GEOPHYSICAL RESEARCH LETTERS, 48.

To Reviewer #2

Reviewer #2 General comments

This paper analyzed lake ice cover and lake water temperature data from the ERA-5 reanalysis and ARC-Lake satellite data to evaluate the trends in ice on and off dates in lakes in the northern hemisphere, as well as the trends in lake surface temperature during the ice on & off months. The authors claim that the warming of lake surface temperature was predominant in the ice-off months compared to that in the ice-on months, that earlier ice off resulted in earlier onset of stratification, that the warming was slower in deeper lakes, and that the predominant warming in the ice-off months could be well explained by earlier ice-off rather than incoming radiations or increasing air temperature.

[Response] We thank the reviewer for the valuable comments and suggestions which have undoubtedly strengthened the analysis presented in our study. Following the reviewer's comments and suggestions, 1) we re-clarified the novelty of our study in understanding the link between earlier lake ice loss and earlier stratification; and 2) we have tried/tested further analyses for the response of the lake-ice system to climate forcing through the intermediary processes, as proposed and inspired by the reviewer. Detailed point-by-point responses are listed below following each comment/suggestion.

Major comments

[Reviewer #2 Major Comment 1]

Significance

1. While the topic is important and has significant impacts on subsequent processes such as lake biogeochemistry. Most of the authors' findings are not new. Earlier onset of stratification and resulting excess warming in lakes are well documented in Woolway et al. (2021), Austin & Coleman (2008, Lake Superior only), as well as loss of ice cover across the lakes in the northern

hemisphere (Sharma et al. 2019), and loss of ice cover in the Great Lakes (Wang et al., 2012; 2018) The slower warming in deeper lakes makes sense and is not a whole new finding.

[Response] We agree with the reviewer that some studies have uncovered individual processes in lake ecosystems including loss of ice cover (in Northern Hemisphere by Magnuson et al., 2000 and Sharma et al., 2019), earlier onset of stratification (in Northern Hemisphere by Woolway et al., 2021 and Kraemer et al., 2015), as well as the linkage between ice cover loss and earlier stratification and its consequent impact on summer lake temperature in individual lakes (in Lake Superior by Austin and Colman 2007, in The Laurentian Great Lakes by Zhong et al., 2016). Yet, our study highlighted that the lake warming rate during the ice-off month is larger than the open-water period (Fig. 1), and the larger warming rate in the ice-off month is mainly induced by the earlier ice loss, rather than change in climate drivers, which have not been reported before. We also explained the drivers for spatial variation in the trend of lake warming during the ice-off month (Fig. 2) and discussed the implication of excess warming due to earlier ice loss on the earlier onset of stratification, which is a novel finding. Following the reviewer's *Major Comment 3*, we have added the explanation of excess lake warming by lake latitude and lake elevation in the revised version, please see details in response to *Reviewer #2 Major Comment 3*. These results also build on the excellent work by Ramon et al. (2021).

[Reviewer #2 Major Comment 2]

2. I felt that these findings around the lake surface warming in responding to ice cover loss provide only a limited snapshot of the much bigger complex picture of the lake-ice systems. The ice loss and the excess warming of the lake surface are intermediary processes. As individual processes, they are relatively well understood. The more important yet unknown is how the lake-ice systems respond to the climate forcing (air temperature, radiation, snowfall on the ice) through all these intermediary processes. I think the author will need to re-work their analyses

in this context. For example, the authors could possibly discuss the frequency of warm winters when lakes were ice free, occurrence of incomplete overturns in dimictic and monomictic lakes, and how they impacted summer stratification overall. Another possibility is to examine a tipping point (if any), such as lake regime change from dimictic to monomictic (e.g., Woolway et al 2020) and cumulative freezing degree days over each lake at which ice cover presented a flexion point in its decrease. They are just suggestions to consider.

[Response] We appreciate the reviewer's valuable comments and suggestions to reinforce our analysis. We agree with the reviewer that given that climate change has influenced lake ice, lake temperature, lake mixing regimes, etc. (Woolway et al., 2020), we should dive deeper into the response of lake-ice systems to climate forcing after finding excess lake warming due to earlier ice-off dates. Following the reviewer's suggestions, we first calculated annual ice-off dates, annual cumulative freezing degree days, and average freezing temperature before ice break-up for each lake grid cell from ERA5 and found their tipping points (if exist) during the study period. We found that the tipping points of ice-off dates are existing in 22% of lake grid cells. While 37% and 43% of lake grid cells exist tipping points of cumulative freezing degree days and average freezing temperature before ice break-up, respectively (Fig. R1a-c). Among the lake grid cells with tipping points for cumulative freezing degree days and average freezing temperature before ice break-up, more than 84% of these lake grid cells have experienced accelerated warming rate after tipping points (Fig. R1e, f, h, i). The spatial patterns of the tipping points for the ice-off date and the cumulative freezing degree days (or average freezing temperature before ice-off dates) are different (Fig. R1a-c). This implies that the accelerated air warming rates during the ice cover duration have not yet approached the threshold or tipping point of lake ice-off date over the study period.

Fig. R1. **a-c**, Spatial patterns of the year when tipping points of ice-off dates (a), cumulative freezing degree days (b), and average freezing temperature before ice-off dates (c) occur. **d-f**, Spatial patterns of the trend in ice-off date (d), cumulative freezing degree days (e), and average freezing temperature before ice-off dates (f) before tipping points, as well as after tipping points (**g**)-(i).

Next, we followed the methods of Woolway and Merchant (2019) and classified lake grid cells

from ERA5 into five mixing regimes, including dimictic, warm monomictic, warm polymictic (the sum of continuous and discontinuous warm polymictic lakes), cold monomictic, and cold polymictic lakes (the sum of continuous and discontinuous cold polymictic lakes). For each lake grid cell from ERA5, we determined the mixing regimes for each year and then determined the dominant lake mixing regimes during the period 1979-1999 and 2000-2020. Fig. R2 shows that > 77% of lakes are dimictic lakes during the two periods, and only < 10% of the lakes experience the shift of dominated mixing regime. To quantify the shift of lake mixing regime from 1979 to 2020, if the frequency of the mixing regime of a lake increase from < 50% to > 80%, or decrease from > 80% to < 50% during the period 2000-2020 relative to the period 1979-1999, this lake is accounted as mixing regime shift. There are only 1,496 lake grid cells (< 2%) experiencing a change in lake mixing regime during the two periods (Fig. R3b, c). More lakes could likely experience the change in lake mixing regime under future climate scenarios (Woolway and Merchant 2019), which is out of the scope of this study. Thus, we did not add the analysis of the change in the lake mixing regime in the revised version.

Fig. R2. Spatial pattern of the dominant lake mixing regimes (including dimictic, warm monomictic, warm polymictic, cold monomictic, and cold polymictic lakes) during the period 1979-1999 (a) and 2000-2020 (b).

Fig. R3. Probability density function (PDF) of change in frequency of lake mixing during the period 2000-2020 relative to the period 1979-1999 (a). Spatial pattern of lakes with decreasing frequency of lake mixing regimes from >80% to <50% (b) and increasing frequency of lake mixing regimes from <50% to >80% (c) during the period 2000-2020 relative to the period 1979-1999.

Following the reviewer’s suggestion, we have analyzed the occurrence of warm winter or incomplete overturn across lakes from ERA5 and how they impact summer stratification. We defined a lake experiencing warm winter (i.e., an ice-free winter) as a lake that warms above 1 °C within a year. A lake experiencing incomplete overturn within a year is defined as a lake with a < 3-day spring mixing duration (the number of days between the ice-off date and onset of stratification) (Pilla and Williamson 2022). There are only 9,063 (8%) lakes in ERA5 experiencing warm winter at least once during the study period and 15,342 lakes (14%) experiencing incomplete overturn at least once during the study period. These small fractions imply that the global-scale response of the lake-ice system to the climate forcing might not be shown through the occurrences of warm winter or incomplete overturn over the study period (Fig. R4a, b). We think that considering the role of warm winter and incomplete overturn in the stratification is important when attributing the variation in the onset of stratification, while this is out of the scope of this study. Moreover, we found that trends in lake surface water

temperature during the ice-off month ($LSWT_{IOFF}$) increase with the occurrence of warm winter or incomplete overturns, while the increase is non-significant ($p > 0.1$; Fig. R4c, d). It is likely that future earlier ice-off dates or more warm winter, as well as more incomplete overturns, could enhance excess warming during the ice-off month. Given that the enhancement is not significant during the study period, we added the role of the occurrence of warm winter and incomplete overturn in excess warming rates during the ice-off month in the Discussion section (Lines 205-207 and Lines 219-221).

Fig. R4. a-b, The relationship between the change in the occurrence of warm winter (a) and the occurrence of the incomplete overturn (b) and the change in the onset of stratification during the period 2000-2020 relative to the period 1979-1999. **c-d**, Boxplot summaries of occurrence of warm winter vs. the trend in lake surface water temperature during the ice-off month ($LSWT_{IOFF}$) (c) and of the occurrence of warm incomplete overturn vs. the trend in $LSWT_{IOFF}$ (d). The black line, and bottom and top edges of the green box in c-d indicate the mean, the 25th, and the 75th percentiles, respectively. The black whiskers indicate the minimum and maximum without outliers.

[Reviewer #2 Major Comment 3]

3. *I am slightly surprised that the paper does not offer any insight on how ice duration and excess warming are related to the elevations, sizes, and latitudes of the lakes. Aren't they any significant relationships?*

[Response] Thanks for this valuable comment. Following the reviewer's comments, we have analyzed the relationships between lake ice cover duration and excess warming and the elevations, sizes, and latitudes of lakes across 963 individual lakes from ARC-Lake and 109,405 lake grid cells from ERA5. For individual lakes from ARC-Lake, we found that mean lake ice duration negatively correlates with lake size ($R = -0.18$, $p < 0.001$; Fig. R5a), positively correlates with latitude ($R = 0.69$, $p < 0.001$; Fig. R5b), and negatively correlates with elevation ($R = -0.13$, $p < 0.001$) across the 963 lakes. The negative relationship between ice cover duration and elevation is counter-intuitive. This could result from the negative relationship between lake latitude and elevation ($R = -0.18$, $p < 0.001$). After controlling lake latitude, the partial correlation coefficient between mean ice cover duration and elevation is significantly positive ($R = 0.72$, $p < 0.001$). The scatter plot between ice cover duration and latitude shows two groups of lakes that can be distinguished by elevation (Fig. R5b). Lakes in Tibet Plateau with high elevation experience a longer ice duration than low-altitude lakes in the same latitude (Fig. R5b, e). For lake grid cells in ERA5, since that lake temperature in ERA5 is simulated by the one-dimensional lake model FLake, "lakes" in ERA5 refer to "typical lakes" for each $0.25^\circ \times 0.25^\circ$ longitude-latitude grid cell with a given lake cover fraction. Thus, there is no actual lake size in the reanalysis data. For gridded elevation, we used elevation data from a global terrain elevation map with a spatial resolution of 30 arc seconds (GTOPO30, <https://lta.cr.usgs.gov/GTOPO30/>). Similarly, we found that high-latitude or high-altitude lake grid cells experience a longer ice duration (Fig. R5c).

Regarding the relationship between excess warming and lake elevations, sizes, and latitude, we didn't find high correlation coefficients between them across the 963 individual lakes from

ARC-Lake ($|R| < 0.06$; Table R1). Moreover, the explanations of spatial variation in trends in LSWT during the ice-off and ice-on month ($LSWT_{IOFF}$, $LSWT_{ION}$) and the sensitivity of lake surface temperature to changes in ice-off and ice-on dates (S_{IOFF} , S_{ION}) are lower than 3% (Table R1). For lake grid cells in ERA5, correlation coefficients between excess warming and lake elevations and latitude are higher than the results of individual lakes in ARC-Lake (Table R1). The explanation of the excess warming by lake latitude and elevation is $< 4\%$ and the explanation of the spatial variation in $LSWT_{IOFF}$, $LSWT_{ION}$, S_{IOFF} , and S_{ION} by lake latitude and elevation is $< 13\%$. In the revised version, we revised sentences to explain the ice cover duration and excess lake warming by lake properties (Lines 69-70 and Lines 101-109), added Table R1 as Supplementary Table 1, and added Fig. R5 as Supplementary Fig. 1.

Fig. R5 (also shown as Supplementary Fig. 1). Relationships between lake ice cover and lake size, lake elevation, and lake latitude for individual lakes in ARC-Lake and lake grid cells in ERA5. **a-b**, Relationship of ice cover and lake size (a) and lake latitude (b) for lakes in ARC-

Lake. **c**, Relationship of ice cover and lake latitude for lakes in ERA5. The colors in **b-c** indicate lake elevation. **d-f**, Spatial patterns of lake size (d), lake elevation (e) for lakes in ARC-Lake, and lake elevation for lakes in ERA5 (f).

Table R1 (also shown as Supplementary Table 1). Correlation coefficients between the trend in lake surface water temperature during the ice-off month ($LSWT_{IOFF}$) and the ice-on month ($LSWT_{ION}$), the ratios of $LSWT_{IOFF}$ to lake warming trend during the open-water season (R_{IOFF} and R_{ION}), the sensitivity of lake surface temperature to changes in ice-off dates (S_{IOFF}), and ice-on dates (S_{ION}), and lake latitude, lake size, and lake elevation for individual lakes in ARC-Lake and lake grid cells in ERA5. Statistically significant correlation coefficients at the 99.9% ($p < 0.001$) level are denoted by one asterisk (*).

	Individual lakes in ARC-Lake (n=963)			Lake grid cells in ERA5 (n=109,405)	
	Latitude	Lake size	Lake elevation	Latitude	Lake elevation
Trend in $LSWT_{IOFF}$	0.06	-0.01	0.09	0.30*	-0.24*
Trend in $LSWT_{ION}$	0.16*	-0.02	-0.08	0.37*	-0.26*
R_{IOFF}	-0.03	-0.05	0.03	0.07*	-0.14*
R_{ION}	-0.02	-0.02	-0.04	0.16*	-0.19*
S_{IOFF}	-0.12	0.05	0.07	-0.24*	0.12*
S_{ION}	0.11	0.07	-0.04	0.20*	-0.09*

[Reviewer #2 Major Comment 4]

4. *The slight cooling of the lakes in southern Canada is interesting, but the reason for this is not discussed at all.*

[Response] Thanks for the valuable comment. Following the reviewer’s suggestion, we have investigated the reasons of the slight cooling of lakes in southern Canada (Fig. R6). We found that lakes in southern Canada have experienced a cooling air temperature during both ice-off month and its preceding month, and consequently a later ice-off date which corresponds to slight lake cooling (Fig. R6a-d). The air cooling before ice break-up in southern Canada is

related to the atmospheric response of the increase in stratospheric polar vortex stretching events due to Arctic change (such as sea ice loss and/or increasing snow cover) in fall and early winter (Cohen et al., 2021). In addition, a diming shortwave radiation in the ice-off month found in the middle of southern Canada (Fig. R6f) could enhance the lake cooling over that area. We have added the reason of slight cooling in lakes in southern Canada in the revised version (Lines 119-120) and Fig. R6 as Supplementary Figure 4.

Fig. R6 (also shown as Supplementary Fig. 4). Explanation for slight cooling of lakes in southern Canada. Spatial pattern of the trend in lake surface water temperature during the ice-off month ($LSWT_{IOFF}$) (a), and the trend in ice-off date (b). Spatial pattern of the trend in air

temperature (T_{air}) (c) and downward short-wave radiation (SW_{down}) (e) during the preceding month before the ice-off month, as well as during the ice-off month (d, f).

[Reviewer #2 Major Comment 5]

Validity

5. *The methodology used in the paper is mostly valid, but there are some insufficient descriptions on their analyses. For example, the authors use ‘air temperature’ and other surface meteorological variables to examine their relationship with R_{ION} , R_{IOFF} etc but there is no description of what period of air temperature etc was used (e.g., annual mean, winter value, or ice-on/off months?). Some of them are pointed out in ‘other points’ below.*

[Response] Here, we used air temperature, downward short- and long-wave radiation during the ice-on/off months to examine the relationships between R_{ION} , R_{IOFF} , and meteorological variables. We have clarified the period of each meteorological variable in the revised version following the reviewer’s suggestions (Lines 104-107 and Line 159). For “other points”, we also added more descriptions for our analysis methodology. Following the reviewer’s Other point 1, we specified the “surface water temperature” in the sentences. Following the reviewer’s Other point 13, we added the description of S_{IOFF} and S_{ION} in the revised Methods. Please see details in response to *Reviewer #2 Other point 1* and *Reviewer #2 Other point 13*.

[Reviewer #2 Major Comment 6]

6. *I am concerned with the author’s statement in the abstract that ‘Notably, the calculated trend in the date of ice loss explains 37% of the spatial variation in excess lake warming, which is greater than that explained by air temperature and downward short and long-wave radiation (< 11%)’. Similar expressions were found multiple times throughout the manuscript. Because ice melting and breakups are driven by incoming radiation and warm air above, they are not*

independent forcings, so the statement could be misleading.

[Response] Thanks for the reviewer's comments. We acknowledge that there are interactions between climate variables on lake warming (For example, long-wave radiation strongly correlates with air temperature). Here, the individual explanation of air temperature, downward short- and long-wave radiation on spatial variation of excess warming has been summarized (the explanation of each variable is less than 11%), NOT the total explanation by the three meteorological variables including their interactions. To make this point clear, the sentence should be read as **“Notably, the calculated trend in the date of ice loss explains 37% of the spatial variation in excess lake warming, which is greater than that explained by air temperature (6%), or downward short- (0%) or long-wave radiation (10%).”** We have clarified in the main text. But due to 150 words limitation for the abstract, this sentence is deleted to fulfill the words accounting.

Other points

[Reviewer #2 Other point 1]

L15: “in the seasonality of lake ice to surface temperature”. Please specify surface air temperature or surface water/ice temperature.

[Response] Here, surface temperature refers to “surface water temperature”. We revised the sentence to **“in the seasonality of lake ice to surface water temperature”**.

[Reviewer #2 Other point 2]

L24: “Excess lake warming during autumn/winter is also sensitive to the date of ice formation, but with a weaker magnitude (0.06 C day-1) due to less incoming radiation.” This sentence seems to imply a cause-and-result relationship but it is unlikely that ice formation timing resulted in warming in preceding autumn/winter.

[Response] Thanks for the reviewer's comments, we agree with the reviewer that lake warming preceding autumn/winter results in the delay of ice formation. As a result, we removed this sentence to avoid confusion.

[Reviewer #2 Other point 3]

L26: "Until the permanent loss of lake ice this century, excess lake warming may be further amplified due to projected future alterations in lake ice phenology." Ice loss is an important factor but another critical point is when warming reaches the point where overturns no longer occur. I think it's worth noting this point.

[Response] Thanks for the comment. We agree with the reviewer that an occurrence of incomplete overturn could be a critical point and enhance lake warming after ice break-up. Following the response to *Reviewer #2 Major Comment 2*, we found enhancement is not significant during the study period across lakes experiencing incomplete overturns (only 14%; Fig. R4d). Thus, we did not revise the sentence here, but revised the Discussion section to consider the role of the occurrence of incomplete overturn in excess lake warming (Lines 219-221).

[Reviewer #2 Other point 4]

L77: Fig. 1b is referred before Fig. 1a.

[Response] Thanks for the reminder. We corrected it in the revised version.

[Reviewer #2 Other point 5]

L81: Unsure why the three selected lakes are representative.... why not means of more lakes over the region?

[Response] Thanks for the comments. We picked one lake for each continent to illustrate our

approach for identifying excess warming and we also have results of means of more lakes over regions shown in Fig. 1d (also shown as Fig. R7). Regarding the reviewer’s comment, we changed the figure legend in Fig. 1a (also shown as Fig. R7) and also added “**in the two lakes located in Europe and North America, but not the one in Asia**” in this paragraph (Line 87) to avoid confusion.

Fig. R7. (also shown as Fig. 1) Excess lake warming during the ice-off or ice-on month. **a**, Trends in monthly lake surface water temperature (LSWT) from 1979 to 2020 across three lakes in Europe, Asia, and North America, respectively. The location of the three lakes is shown as black dots in **b**, **c**. The dashed lines and shaded areas represent the mean and range of ice-off date or ice-on date from 1979 to 2020. **b-c**, Spatial pattern of mean ice-off month (**b**), ice-on month (**c**). **d**, Mean trend in monthly LSWT from 1979 to 2020 across lakes with the same ice-off month from March to July. **e-f**, Spatial pattern of ratios of the trend in LSWT during the ice-off month ($LSWT_{IOFF}$) or LSWT during the ice-on month ($LSWT_{ION}$) to the trend in LSWT during the open-water period (R_{IOFF} (**e**) and R_{ION} (**f**)). The inset in **b-c**, **e-f** shows the probability density function (PDF) from Europe (orange), North America (pink), and Asia (blue),

respectively.

[Reviewer #2 Other point 6]

L85: “The lake surface temperature trend.... (Fig. 1a)”. It seems inadequate to draw conclusions for the entire NA, Europe or Asia based on the 3 lakes only.

[Response] Following the response to *Reviewer #2. Other point 5*, we changed the figure legend in Fig. 1a (also shown as Fig. R7) and also added **“in the two lakes located in Europe and North America, but not the one in Asia”** in this paragraph (Line 87) to avoid confusion.

[Reviewer #2 Other point 7]

L94: For R_{ION} , R_{IOFF} calculation, how were the open water seasons defined? unique to each lake and year or constant across the lakes and the study period?

[Response] Open-water seasons for each lake in our study are defined as months after the ice-off month and before the ice-on month. For each lake, the ice-off / ice-on month is defined as the month of mean ice-off / ice-on date, NOT varies between years, i.e. constant over the study period 1979-2020 (Lines 382-385). Thus, the open-water season for each lake is constant over the study period. As the ice-off and ice-on dates are different across lake grid cells, the open-water season varies across the 109,405 lake grid cells.

[Reviewer #2 Other point 8]

L132: “We propose that the excess warming observed in the studied lakes is driven primarily by additional radiation input due to the advancement of ice break-up under climate change.” This is a well-known ice-albedo feedforward process and not really a new proposal.

[Response] Thanks for the reminder. We have deleted “We propose that” and added one reference (Austin and Colman 2007) for the well-known ice-albedo feedforward process.

[Reviewer #2 Other point 9]

L172: “The difference between S_{IOFF} and S_{ION} across the...” Also, water temperature has a lower limit at the freezing degree, so below this point, it is insensitive no matter how cold the air above is.

[Response] S_{IOFF} is the ratio of the trend in LSWT_{IOFF} to the trend in ice-off date and S_{ION} is the ratio of the trend in LSWT_{ION} to the trend in ice-on date. Thus, S_{IOFF} / S_{ION} represents the lake warming rate due to extra incoming radiation into lake water resulting from one-day earlier ice break-up / later ice formation. The incoming radiation during the ice-off month and ice-on month could partly explain the differences between S_{IOFF} and S_{ION}.

[Reviewer #2 Other point 10]

ERA-5 lake data: What’s the minimum size of lakes covered in this study? If more than one lake is included in a 0.25degx0.25deg cell, are they considered as one lake consolidated or is there any subgrid scale treatments?

[Response] For lake grid cells in ERA5, since that lake temperature in ERA5 is simulated by the one-dimensional lake model FLake, and only one vertical profile lake water temperature is simulated for one lake grid cell, that refers to “a typical lake” for each 0.25° x 0.25° longitude-latitude grid cell. Thus, there is no subgrid scale treatment in ERA5 lake data and there is no lake size for the “typical lake” in ERA5 lake data.

[Reviewer #2. Other point 11]

L300: “identified ice-cover periods when lake surface temperature is less than 1 C”. Why do you use the lake surface temperature to determine an ice-covered period when you have ice cover data? Why not use the ice cover data directly?

[Response] Thanks for the suggestions. We have used both lake ice thickness and lake surface temperature to determine the ice-covered period. As shown in Supplementary Figs. 16-17, we found annual ice-covered periods derived from ERA5 ice cover data show earlier annual ice-on dates and later annual ice-off dates than that derived from ERA5 lake surface temperature (Supplementary Figure 16) and derived from AMSR-E/2 (Supplementary Figure 17). This may be related to missing or under-represented ice-related processes in FLake, e.g., the absence of the snow model in FLake, while snow cover could have higher albedo and greater insulation effects on ice formation or breakup (Sharma et al., 2020). The later annual ice-off dates may be also related to the absence of heating of water by solar radiation penetrating down the ice cover (Su et al., 2019). We also evaluated the lake ice phenology derived from both ERA5 lake surface temperature (Supplementary Figure 13) and ERA5 ice cover data (Supplementary Figure 17) with AMSR-E/2 observations. For ice-off dates, results derived from both ERA5 lake surface temperature and ice cover show great agreement with AMSR-E/2 ice-off dates ($R=0.84-0.85$). The RMSE between ice-off dates derived from ERA5 lake surface temperature and ice cover and satellite ice-off dates are 18.2 and 17.8 days, respectively. For ice-on dates, the RMSE between results from ERA5 lake surface temperature and ice cover and AMSR-E/2 are 24.4 and 32.2 days, respectively. Given that the ice-on date derived from ERA5 lake surface data agrees better than that derived from ERA5 lake ice data with AMSR-E/2 data, we used the lake surface temperature to determine an ice-covered period in the main text, and results from ERA5 lake ice depth are shown in Supplementary Figure 18.

[Reviewer #2. Other point 12]

L306: *“Meanwhile, we excluded lakes with an ice-cover duration < 30 days to prevent lakes experiencing break-up and freeze-up in the same month (< 0.2%).” What’s the fraction of data removed by this criteria?*

[Response] For individual lakes in ARC-Lake, there are 9 lakes (0.1%) removed by this criteria. For lake grid cells in ERA5, there are 282 lake grid cells (0.25%) removed by this criteria. We revised the sentence as “**Meanwhile, we excluded lakes with an ice-cover duration < 30 days to prevent lakes experiencing break-up and freeze-up in the same month (< 0.2% of lakes in ARC-Lake and <0.3% of lake grid cells in ERA5).**”

[Reviewer #2. Other point 13]

Calculation methods of S_{ION} , S_{IOFF} , the product of the trend in ice-off date by mean downward short- and long-wave radiation during the ice-off month are not described in Methods. For S_{ION} & S_{IOFF} , the main text says “See Methods”.

[Response] Thanks for the reminder. Following the response of *Reviewer #2. Other point 2*, we have revised the Methods section and added a clear description of the S_{ION} , S_{IOFF} (Lines 378-380).

[Reviewer #2. Other point 14]

Figure 2: This figure appears very crowded. b.c.g.h are really hard to see. I understand that there is a limitation in the figure numbers. However, I encourage the authors to make an effort to make it more concise

[Response] Thanks for this comment. Following the reviewer’s suggestions, in the revised version, we used Fig.2 and Fig.3 to show the explanation for excess lake warming during the ice-off and ice-on month, respectively, which is also shown in Fig. R8 and Fig. R9.

Fig. R8. (also shown as **Fig. 2**) Explanation for excess warming in the ice-off month. **a-c**, Spatial pattern of the trend in LSWT_{IOFF} (a), ice-off date (b), and lake depth (c). **d-e**, Relationship between the trend in LSWT_{IOFF} (y-axis) and the trend in ice-off date (d) and lake depth (e) (x-axis). The color of points in d-e shows the number of lakes.

Fig. R9. (also shown as **Fig. 3**) Explanation for excess warming in the ice-on month. **a-c**, Spatial pattern of the trend in LSWT_{ION} (a), ice-on date (b), and downward longwave radiation (LWdown) in the ice-on month (c). **d-e**, Relationship between the trend in LSWT_{ION} (y-axis) and the trend in ice-on date (d) and downward LWdown (e) in the ice-on month (x-axis). The color of points in d-e shows the number of lakes.

References

- Austin, J. A. & S. M. Colman (2007) Lake Superior summer water temperatures are increasing more rapidly than regional air temperatures: A positive ice-albedo feedback. *GEOPHYSICAL RESEARCH LETTERS*, 34.
- Cohen, J., et al. (2021) Linking Arctic variability and change with extreme winter weather in the United States. *SCIENCE*, 373, 1116-1121.
- Kraemer, B. M., et al. (2015) Morphometry and average temperature affect lake stratification responses to climate change. *GEOPHYSICAL RESEARCH LETTERS*, 42, 4981-4988.
- Magnuson, J. J., et al. (2000) Historical trends in lake and river ice cover in the Northern Hemisphere. *SCIENCE*, 289, 1743-1746.
- Pilla, R. M. & C. E. Williamson (2022) Earlier ice breakup induces change point responses in duration and variability of spring mixing and summer stratification in dimictic lakes. *LIMNOLOGY AND OCEANOGRAPHY*, 67, S173-S183.
- Ramon, C. L., et al. (2021) Bathymetry and latitude modify lake warming under ice. *HYDROLOGY AND EARTH SYSTEM SCIENCES*, 25, 1813-1825.
- Sharma, S., et al. (2019) Widespread loss of lake ice around the Northern Hemisphere in a warming world. *NATURE CLIMATE CHANGE*, 9, 227-+.
- Sharma, S., et al. (2020) Integrating Perspectives to Understand Lake Ice Dynamics in a Changing World. *JOURNAL OF GEOPHYSICAL RESEARCH-BIOGEOSCIENCES*, 125.
- Su, D. S., et al. (2019) Numerical study on the response of the largest lake in China to climate change. *HYDROLOGY AND EARTH SYSTEM SCIENCES*, 23, 2093-2109.
- Woolway, R. I., et al. (2020) Global lake responses to climate change. *NATURE REVIEWS*

- EARTH & ENVIRONMENT, 1, 388-403.
- Woolway, R. I. & C. J. Merchant (2019) Worldwide alteration of lake mixing regimes in response to climate change. NATURE GEOSCIENCE, 12, 271-+.
- Woolway, R. I., et al. (2021) Phenological shifts in lake stratification under climate change. NATURE COMMUNICATIONS, 12.
- Zhong, Y. F., et al. (2016) Recent accelerated warming of the Laurentian Great Lakes: Physical drivers. LIMNOLOGY AND OCEANOGRAPHY, 61, 1762-1786.

Reviewers' Comments:

Reviewer #1:

Remarks to the Author:

Dear Authors,

I appreciate the work made to improve the quality of the manuscript. I feel comfortable with the answers provided to both reviewers. I have only two remarks in the new version of the manuscript:

L219-221. How do the authors know whether there was an incomplete overturn? How do the authors define the onset of stratification? A full overturn might have happened before the ice formation. Could the authors add the method used to characterise the onset of stratification in the method section? It would be helpful to illustrate an example in the supplementary material showing a "well-mixed temperature profile" and a "stratified temperature profile" at the onset condition. There are different ways to do the above; it would be good if the authors provided the approach they used.

L261. I agree that a horizontally-averaged [vertical] temperature profile represents lake-mean conditions. I think that the end of the sentence "we elected to investigate simulated temperatures from a one-dimensional (1D) lake model, which also largely represents lake-mean conditions." requires a reference that addresses and supports this statement. If not, I recommend removing "which also largely represents lake-mean conditions". The papers by Ulloa et al. (2019 GRL) and Ramón et al. (2021 HESS) discuss precisely this matter. They show that topography and flow three-dimensionality affect the background temperature distribution in ice-covered lakes. In the case of open water bodies, one may consider that wind supports lateral homogenisation of the background temperature structure. However, wind also supports 1) lateral heterogeneities in vertical mixing rates, especially between the littoral and pelagic regions, and, eventually, 2) persistent horizontal gradients in the temperature field.

All the best,
Hugo N Ulloa

Reviewer #2:

Remarks to the Author:

Review of "Earlier ice loss accelerates lake warming in the Northern Hemisphere" by Xinyu Li, Shushi Peng, Yi Xi, R. Iestyn Woolway, Gang Liu.

This is my second review of the manuscript. The authors did a nice work to improve the manuscript to address the comments by myself and the other reviewers. I really appreciate the authors' efforts to make additional analyses, some of which are now included in the manuscript.

This includes the newly added analysis on relationships between ice duration and lake size, elevation, and latitude are valuable. The newly added section on the limitation of the study is also useful. I also think the methodology section is greatly improved.

While the manuscript can be of great importance to the scientific community, I still have two relatively major issues:

Readability issues: The manuscript is overall well written. However, some parts of the manuscript are really hard to understand and I had to read multiple times to get the idea. I listed examples in "Other points" below. I also saw some mis-referring the figures and typos. I suggest having the manuscript proofread carefully by all co-authors and perhaps a professional editor to fix the errors and improve the flow.

Use of "Explanation": I see expressions like "spatial variation of X can be explained by Y at Z%..." in some places (examples given in "Other points"). In almost all places, I do not see how the percentages were calculated. In addition, figures and tables referred to in such expressions are about simply correlation coefficients. If the authors state something based on correlation coefficients, I suggest simply saying "X correlates with Y" but not "X is explained by Y". Because just a significant correlation does not necessarily guarantee cause-and-result or physically meaningful relationships. "Explain" is indicative of such relationships. If the authors would like to indicate such relationships, I'd suggest an expression like "X correlates with Y, indicating Y explains X through mechanism A,B, and C"

Other points:

L18: "Until the permanent loss of lake ice this century," I don't understand what the authors mean by this statement.... In climate projections, lake ice will decline and some will lose ice cover completely by the end of the century but not all of them (e.g., Xiao et al. 2015; Woolway and Merchant, 2019). Perhaps this should be "Until permanent loss of lake ice in the future,"?

Xiao, C.; Lofgren, B.M.; Wang, J.; Chu, P.Y. A Dynamical Downscaling Projection of Future Climate Change in the Laurentian Great Lakes Region Using a Coupled Air-Lake Model. Preprints 2018, 2018070468 (doi: 10.20944/preprints201807.0468.v1).

L23, L60, L342, L397: I see notations of "ref" before citation numbers. Are they typos?

L53 "in-situ" should be "in situ" (Latin phrases do not need a hyphen when used as an adjective. It should still be italicized.)

L75: "by selecting three ice-covered lakes...". Are their elevations similar?

L92: "the average RIOFF of lakes in the Northern Hemisphere increases from 1.2 ± 0.9 to 1.4 ± 1.3 (\pm standard deviation across Northern Hemisphere lakes; Fig. 1d)." Fig. 1d is not about R_IOFF

L101, 105, 158, 198: add "surface" before "air temperature".

L198: Add "the" before "Northern Hemisphere".

L104-106: "However, our analysis suggests that air temperature, downward short- and long-wave radiation during the ice-off month only explain 4%, 0%, and 5%, respectively, of the spatial variation in RIOFF" Where are the numbers of 4%, 0%, and 5% coming from? I do not see them in Supplementary Figure 3. The same comment goes to the sentence right after.

Supplementary Figure 3, 19: I suggest not cross referring the figures on the main manuscript (e.g., "Same as Fig. 2 and Fig. 3"). It's confusing.

Supplementary Figure 3: "e-g, Relationships between RIOFF and Tair (e), RIOFF and SWdown (f), and RIOFF and LWdown (g)." Shouldn't it be "Relationships between RIOFF and the trends of Tair (e), RIOFF and SWdown (f), and RIOFF and LWdown (g)"?

L108: "In addition, the explanation of spatial variation in R_IOFF and R_ION by latitude and elevation is less than 4% (Supplementary Table 1)." I don't understand this sentence. Supplementary Table 1 simply provides correlation coefficients. Shouldn't this be just talking about correlations? Also, where is the 4% coming from? I don't see it in Supplementary Table 1.

L122: "As well as the advancement of ice-off, the extra incoming radiation into lakes also depends on the absolute downward short- and long-wave radiation at that time of year. Therefore, we used the product of..." These sentences are hard to understand... How about this: "In addition to the advancement of ice-off (or earlier ice melting), the seasonality of downward short- and long-wave radiations also contributes to the increase in incoming radiation into lakes. In order to evaluate the net increase in incoming radiation due to these factors, we used the product of..."

L126: "... indicate extra incoming radiation (Supplementary Fig. 4)." Supplementary Figure 4 is not about ΔE_{IOFF_SW} etc.

L127: "(+3%, +5% for downward short- and long-wave radiation)". I don't see these numbers on Supplemental Figure 5. Please state how these numbers are calculated.

L162: "Sensitivity of excess lake warming to changes in ice phenology". Could you touch on how lake elevation plays a role in the sensitivity of S_{IOFF} and S_{IOFF} ? It seems natural to mention as the earlier analysis includes lake elevation.

L217: "Fifteen percent..." why is a word expression used here while numerical expressions are used in other places?

L246, 262: Add "water" before "temperature"

L259: Suggest citing a couple of large lake studies here. See examples below:

Fujisaki-Manome, A. et al. Simulating Impacts of Precipitation on Ice Cover and Surface Water Temperature Across Large Lakes. *Journal of Geophysical Research: Oceans* 125, 1–18 (2020).

Ye, X., Anderson, E. J., Chu, P. Y., Huang, C. & Xue, P. Impact of Water Mixing and Ice Formation on the Warming of Lake Superior: A Model-guided Mechanism Study. *Limnology and Oceanography* 64, 558–574 (2019).

Xue, P. et al. Improving the simulation of large lakes in regional climate modeling: Two-way lake-atmosphere coupling with a 3D hydrodynamic model of the great lakes. *Journal of Climate* 30, 1605–1627 (2017).

L285: "during the months ice-off" should be "during the ice-off months"

L287: "This sensitivity is mainly influenced by lake depth and incoming radiation."

Response to the reviewers

To Reviewer #1

Reviewer #1 (Remarks to the Author):

Dear Authors,

I appreciate the work made to improve the quality of the manuscript. I feel comfortable with the answers provided to both reviewers. I have only two remarks in the new version of the manuscript.

[Response] We thank the reviewer again for the positive feedback. Following the reviewer's comments and suggestions, we have further revised the manuscript and now included definitions of incomplete overturn and onset of stratification (as requested).

[Reviewer #1 General Comment 1]

L219-221. How do the authors know whether there was an incomplete overturn? How do the authors define the onset of stratification? A full overturn might have happened before the ice formation. Could the authors add the method used to characterise the onset of stratification in the method section? It would be helpful to illustrate an example in the supplementary material showing a "well-mixed temperature profile" and a "stratified temperature profile" at the onset condition. There are different ways to do the above; it would be good if the authors provided the approach they used.

[Response] Thank you for the valuable suggestion. In our study, the onset of stratification is defined as the first day when the temperature difference between the epilimnion and the hypolimnion is greater than 1 °C and lasts for three consecutive days (Woolway et al. 2014). In this way, incomplete overturn occurs when lake's mixing duration (the number of days between the ice-off date and onset of stratification) is < 3 days (Pilla and Williamson 2022). Given that our study focuses on excess warming after ice break-up, we only consider ice break-up, overturn and stratification related to ice break-up. We agree with the reviewer that complete or incomplete overturn could also occur in autumn, i.e., before the ice formation, but it is out of the scope of this study. We have added the definitions of the onset of stratification and incomplete overturn in the Methods (Lines 400-404, also copied as below) and a schematic of "well-mixed temperature profile" and "stratified temperature profile" in the supplementary material (Fig. R1, also added as Supplementary Fig. 20), which should be useful to the broad readership of the journal.

“The onset of stratification is defined as the first day when the temperature difference between the epilimnion and the hypolimnion is greater than 1 °C and lasts for three consecutive days (Supplementary Fig. 20). The occurrence of incomplete overturn of a lake is defined as when the lake’s mixing duration (the number of days between the ice-off date and onset of stratification) is < 3 days⁴³.”

Fig. R1 (also added as Supplementary Fig. 20). Schematic of the lake vertical structure and temperature profile during complete mixing (a) and stratification (b).

[Reviewer #1 General Comment 2]

L261. I agree that a horizontally-averaged [vertical] temperature profile represents lake-mean conditions. I think that the end of the sentence “we elected to investigate simulated temperatures from a one-dimensional (1D) lake model, which also largely represents lake-mean conditions.” requires a reference that addresses and supports this statement. If not, I recommend removing “which also largely represents lake-mean conditions”. The papers by Ulloa et al. (2019 GRL) and Ramón et al. (2021 HESS) discuss precisely this matter. They show that topography and flow three-dimensionality affect the background temperature distribution in ice-covered lakes. In the case of open water bodies, one may consider that wind supports lateral homogenisation of the background temperature structure. However, wind also supports 1) lateral heterogeneities in vertical mixing rates, especially between the littoral and pelagic regions, and, eventually, 2) persistent horizontal gradients in the temperature field.

[Response] Thank you for this kind reminder. Following the reviewer’s suggestion, we have removed “which also largely represents lake-mean conditions” in the revised manuscript. We also now cited the papers by Ulloa et al., (2019) and Ramón et al., (2021), which we found insightful.

References

Pilla, R. M. & C. E. Williamson (2022) Earlier ice breakup induces changepoint responses in

duration and variability of spring mixing and summer stratification in dimictic lakes. LIMNOLOGY AND OCEANOGRAPHY, 67, S173-S183.

Woolway, R. I., et al. (2014) A novel method for estimating the onset of thermal stratification in lakes from surface water measurements. WATER RESOURCES RESEARCH, 50, 5131-5140.

Ulloa, H. N., et al. (2019) Differential Heating Drives Downslope Flows that Accelerate Mixed-Layer Warming in Ice-Covered Waters. GEOPHYSICAL RESEARCH LETTERS, 46, 13872-13882.

Ramón, C. L., et al. (2021) Bathymetry and latitude modify lake warming under ice. HYDROLOGY AND EARTH SYSTEM SCIENCES, 25, 1813-1825.

To Reviewer #2

Reviewer #2 General comments

Reviewer #2 (Remarks to the Author):

Review of “Earlier ice loss accelerates lake warming in the Northern Hemisphere” by Xinyu Li, Shushi Peng, Yi Xi, R. Iestyn Woolway, Gang Liu.

This is my second review of the manuscript. The authors did a nice work to improve the manuscript to address the comments by myself and the other reviewers. I really appreciate the authors’ efforts to make additional analyses, some of which are now included in the manuscript. This includes the newly added analysis on relationships between ice duration and lake size, elevation, and latitude are valuable. The newly added section on the limitation of the study is also useful. I also think the methodology section is greatly improved.

While the manuscript can be of great importance to the scientific community, I still have two relatively major issues:

[Response] We thank the reviewer for the positive appraisal of our work. We very much appreciate the detailed comments provided by the reviewer. Following the reviewer’s comments and suggestions, we have carefully checked the manuscript and corrected all grammatical errors as well as addressed any outstanding issues raised by the reviewer. Please see the details in our point-by-point response.

Major issues

[Reviewer #2 Major issue 1]

Readability issues: The manuscript is overall well written. However, some parts of the manuscript are really hard to understand and I had to read multiple times to get the idea. I listed examples in “Other points” below. I also saw some mis-referring the figures and typos. I suggest having the manuscript proofread carefully by all co-authors and perhaps a professional editor to fix the errors and improve the flow.

[Response] Thank you for bringing this to our attention. We have now carefully revised our manuscript, which has also been proofread carefully by all co-authors.

[Reviewer #2 Major issue 2]

Use of “Explanation”: I see expressions like “spatial variation of X can be explained by Y at

Z%..." in some places (examples given in "Other points"). In almost all places, I do not see how the percentages were calculated. In addition, figures and tables referred to in such expressions are about simply correlation coefficients. If the authors state something based on correlation coefficients, I suggest simply saying "X correlates with Y" but not "X is explained by Y". Because just a significant correlation does not necessarily guarantee cause-and-result or physically meaningful relationships. "Explain" is indicative of such relationships. If the authors would like to indicate such relationships, I'd suggest an expression like "X correlates with Y, indicating Y explains X through mechanism A,B, and C".

[Response] Thank you for this comment and suggestion. In this study, we used the coefficient of determination (R^2) to investigate the correlation between different variables, as the reviewer describes above. The R-squared values are quoted as percentage (i.e., the percent explained by the relationship within a linear regression). It is this correlation coefficient that we describe in the figures and tables. To make our manuscript easily understood, we changed the coefficients of determination from correlation coefficients to R-squared in Supplementary Figs. 3, 5, 6 and added results of R-squared in Supplementary Table 1, 2, 4 in the revised version. Please see details in response to *Reviewer #2* Other points 8, 11, and 14.

For the use of "Explain", we agree with the reviewer that correlation coefficients, or R-squared, do not imply causality, which could be expressed as "explain". However, the expressions using "explain" or "explanation" in our study are all indicating the impacts of climate variables, lake ice phenology, or lake depth on the lake warming during the ice-on and ice-off month, these cause-and-result relationships have been emphasized in the sentence in Lines 101-104 (**As potential drivers of lake surface temperature, climatic variables including surface air temperature³¹, downward short-wave radiation²⁴, and downward long-wave radiation³², and lake geographic variables including latitude³³ and elevation³⁴, as well as lake ice phenology²⁶ may explain the across-lake variations in the magnitude of excess lake warming.**). For this reason, we kept using the expression of "X can be explained by Y at Z%" in the revised version, but have included a caveat "**Note that the percentages quoted describe the coefficient of determination, which estimates the percentage of variability that can be explained by a regression model and does not imply causality**" (Lines 107-109).

Other points

[Reviewer #2 Other point 1]

L18: "Until the permanent loss of lake ice this century," I don't understand what the authors mean by this statement.... In climate projections, lake ice will decline and some will lose ice

cover completely by the end of the century but not all of them (e.g., Xiao et al. 2015; Woolway and Merchant, 2019). Perhaps this should be “Until permanent loss of lake ice in the future,”?

Xiao, C.; Lofgren, B.M.; Wang, J.; Chu, P.Y. A Dynamical Downscaling Projection of Future Climate Change in the Laurentian Great Lakes Region Using a Coupled Air-Lake Model. Preprints 2018, 2018070468 (doi: 10.20944/preprints201807.0468.v1).

[Response] Thank you. Following the reviewer’s suggestion, we revised this sentence to “**Until the permanent loss of lake ice in the future, excess lake warming may be further amplified due to projected future alterations in lake ice phenology.**” We also now include a reference for Xiao et al. in the manuscript (Line 287).

[Reviewer #2 Other point 2]

L23, L60, L342, L397: I see notations of “ref” before citation numbers. Are they typos?

[Response] No, they are used following journal citation style. We used “ref” here to avoid confusion from directly connecting numbers (Line 60), or “et al.” (Line 344, Line 399) or unit (°N in Line 23, ° in Line 301) with citation numbers, following the guideline of citation style from the journal. This will be addressed by the journal if accepted.

[Reviewer #2 Other point 3]

L53” “in-situ” should be “in situ” (Latin phrases do not need a hyphen when used as an adjective. It should still be italicized.).

[Response] It has been corrected.

[Reviewer #2 Other point 4]

L75: “by selecting three ice-covered lakes...”. Are their elevations similar?

[Response] The three ice-covered lakes are all situated at low elevations (49 m, 65 m and 164 m). This information is now included in the manuscript (Line 76).

[Reviewer #2 Other point 5]

L92: “the average RIOFF of lakes in the Northern Hemisphere increases from 1.2 ± 0.9 to 1.4 ± 1.3 (\pm standard deviation across Northern Hemisphere lakes; Fig. 1d).” Fig. 1d is not about R_{IOFF} .

[Response] With “With the ice-off month from March to July” shown in this sentence before, R_{IOFF} for each ice-off month can be derived from Fig. 1d, thus we referred to Fig. 1d here. We

understand the reviewer's comment, and we have now removed Fig. 1d from this sentence.

[Reviewer #2 Other point 6]

L101, 105, 158, 198: add "surface" before "air temperature".

[Response] It has been added.

[Reviewer #2 Other point 7]

L198: Add "the" before "Northern Hemisphere".

[Response] It has been added.

[Reviewer #2 Other point 8]

L104-106: "However, our analysis suggests that air temperature, downward short- and long-wave radiation during the ice-off month only explain 4%, 0%, and 5%, respectively, of the spatial variation in R_{IOFF} " Where are the numbers of 4%, 0%, and 5% coming from? I do not see them in Supplementary Figure 3. The same comment goes to the sentence right after.

[Response] Thank you for this query. The numbers of 4%, 0% and 5% represent the R-squared derived from the correlation analysis. Following the response to *Reviewer #2 Major issue 2*, we added the results of R-squared to show the correlation shown in Supplementary Fig. 3. In addition, to make the expression of the percentages clear, we revised the sentence as "**However, our analysis suggests that surface air temperature, downward short- and long-wave radiation during the ice-off month only explain 4%, 0%, and 5%, respectively, of the spatial variation in R_{IOFF} (Supplementary Fig. 3; Note that the percentages quoted describe the coefficient of determination, which estimates the percentage of variability that can be explained by a regression model and does not imply causality).**".

[Reviewer #2 Other point 9]

Supplementary Figure 3, 19: I suggest not cross referring the figures on the main manuscript (e.g., "Same as Fig. 2 and Fig. 3"). It's confusing.

[Response] We have removed all the cross referring in the revised version, and wrote the figure captions independently for each figure.

[Reviewer #2 Other point 10]

Supplementary Figure 3: "e-g, Relationships between R_{IOFF} and T_{air} (e), R_{IOFF} and SW_{down} (f), and R_{IOFF} and LW_{down} (g)." Shouldn't it be "Relationships between R_{IOFF} and

the trends of Tair (e), RIOFF and SWdown (f), and RIOFF and LWdown (g)”?

[Response] It has been corrected.

[Reviewer #2. Other point 11]

L108: “In addition, the explanation of spatial variation in R_{IOFF} and R_{ION} by latitude and elevation is less than 4% (Supplementary Table 1).” I don’t understand this sentence. Supplementary Table 1 simply provides correlation coefficients. Shouldn’t this be just talking about correlations? Also, where is the 4% coming from? I don’t see it in Supplementary Table 1.

[Response] Thank you. This sentence is indeed referring to R-squared. Following the response to Reviewer #2 Major issue 2, we added the results of R-squared in Table 1, which is also shown in Table R1. Because R-squared between R_{IOFF} and R_{ION} and latitude and elevation ranges from 0.00 to 0.03, we used $< 4\%$ in the main text.

Table R1 (also shown as Supplementary Table 1). Correlation coefficients (R-squared) between the trend in lake surface water temperature during the ice-off month ($LSWT_{IOFF}$) and the ice-on month ($LSWT_{ION}$), the ratios of $LSWT_{IOFF}$ to lake warming trend during the open-water season (R_{IOFF} and R_{ION}), the sensitivity of lake surface temperature to changes in ice-off dates (S_{IOFF}), and ice-on dates (S_{ION}), and lake latitude, lake size, and lake elevation for individual lakes in ARC-Lake and lake grid cells in ERA5. Statistically significant correlation coefficients at the 99.9% ($p < 0.001$) level are denoted by one asterisk (*), with R-squared in parentheses.

	Individual lakes in ARC-Lake (n = 963)			Lake grid cells in ERA5 (n = 109,405)	
	Latitude	Lake size	Lake elevation	Latitude	Lake elevation
Trend in $LSWT_{IOFF}$	0.06(0.00)	-0.01(0.00)	0.09(0.01)	0.30(0.09)*	-0.24(0.06)*
Trend in $LSWT_{ION}$	0.16(0.03)*	-0.02(0.00)	-0.08(0.01)	0.37(0.14)*	-0.26(0.07)*
R_{IOFF}	-0.03(0.00)	-0.05(0.00)	0.03(0.00)	0.07(0.00)*	-0.14(0.02)*
R_{ION}	-0.02(0.00)	-0.02(0.00)	-0.04(0.00)	0.16(0.02)*	-0.19(0.03)*
S_{IOFF}	-0.12(0.01)	0.05(0.00)	0.07(0.00)	-0.24(0.06)*	0.12(0.01)*
S_{ION}	0.11(0.01)	0.07(0.00)	-0.04(0.00)	0.20(0.04)*	-0.09(0.01)*

[Reviewer #2. Other point 12]

L122: *“As well as the advancement of ice-off, the extra incoming radiation into lakes also depends on the absolute downward short- and long-wave radiation at that time of year. Therefore, we used the product of...”* These sentences are hard to understand.... How about this: *“In addition to the advancement of ice-off (or earlier ice melting), the seasonality of downward short- and long-wave radiations also contributes to the increase in incoming radiation into lakes. In order to evaluate the net increase in incoming radiation due to these factors, we used the product of...”*

[Response] Thank you. we have revised this sentence following the reviewer’s suggestion.

[Reviewer #2. Other point 13]

L126: *“... indicate extra incoming radiation (Supplementary Fig. 4).”* Supplementary Figure 4 is not about ΔE_{IOFF_SW} etc.

[Response] Thank you. We have revised this sentence to **“... indicate extra incoming radiation (Supplementary Fig. 5).”**

[Reviewer #2. Other point 14]

L127: *“(+3%, +5% for downward short- and long-wave radiation) “.* I don’t see these numbers on Supplemental Figure 5. Please state how these numbers are calculated.

[Response] Thank you. Following the response to Reviewer #2 Major issue 2, we showed the results of R-squared in Supplementary Fig. 5. Here, the +3% (we revised it to +4%) represents the difference between R-squared shown in Supplemental Fig. 5f and in Supplemental Fig. 5e, and +5% (we revised it to +6%) represents the difference between Supplemental Fig. 5g and Supplemental Fig. 5e.

[Reviewer #2. Other point 15]

L162: *“Sensitivity of excess lake warming to changes in ice phenology”.* Could you touch on how lake elevation plays a role in the sensitivity of S_{IOFF} and S_{ION} ? It seems natural to mention as the earlier analysis includes lake elevation.

[Response] In the previous version, Supplementary Table 1 shows the correlation coefficients between lake elevation and sensitivity of excess lake warming to changes in ice phenology ($R = 0.12$ and $R = -0.09$ between lake elevation and S_{IOFF} and S_{ION} respectively). In the revised version, we added one sentence for correlation between lake elevation and S_{IOFF} (S_{ION}) here

(Lines 196-197).

[Reviewer #2. Other point 16]

L217: “Fifteen percent....” why is a word expression used here while numerical expressions are used in other places?

[Response] We have revised this sentence by using numerical expressions.

[Reviewer #2. Other point 17]

L246, 262: Add “water” before “temperature.

[Response] We have added it in the revised version.

[Reviewer #2. Other point 18]

L259: Suggest citing a couple of large lake studies here. See examples below:

Fujisaki-Manome, A. et al. Simulating Impacts of Precipitation on Ice Cover and Surface Water Temperature Across Large Lakes. Journal of Geophysical Research: Oceans 125, 1–18 (2020).

Ye, X., Anderson, E. J., Chu, P. Y., Huang, C. & Xue, P. Impact of Water Mixing and Ice Formation on the Warming of Lake Superior: A Model-guided Mechanism Study. Limnology and Oceanography 64, 558–574 (2019).

Xue, P. et al. Improving the simulation of large lakes in regional climate modeling: Two-way lake-atmosphere coupling with a 3D hydrodynamic model of the great lakes. Journal of Climate 30, 1605–1627 (2017).

[Response] We have added these citations following the reviewer’s suggestion.

[Reviewer #2. Other point 19]

L285: “during the months ice-off” should be “during the ice-off months”.

[Response] It has been corrected.

[Reviewer #2. Other point 20]

L287: “This sensitivity is mainly influenced by lake depth and incoming radiation.”

[Response] This sentence has been deleted in the revised version.